



# NorESM2–DIAM: A coupled model for investigating global and regional climate-economy interactions

Jenny Bjordal[1], Anthony A. Smith, Jr.[2,3], Henri Cornec[2], and Trude Storelvmo[1,4]

[1]Department of Geosciences, University of Oslo, Oslo, Norway
[2]Department of Economics, Yale University, New Haven, CT, United States of America
[3]National Bureau of Economic Research, Cambridge, MA, United States of America
[4]Nord University Business School, Bodø, Norway

**Correspondence:** Jenny Bjordal (jenny.bjordal@geo.uio.no)

**Abstract.** Global warming poses substantial risks to natural and human systems worldwide. Understanding the complex interactions between climate change and the economy is essential for designing effective policies and mitigation strategies. Yet, existing modeling tools are often limited by coarse spatial aggregation, simplified climate representation, or lack of interaction between climate and the economy. To address these gaps, we develop a novel framework that couples an Earth System

Model (ESM)—the Norwegian Earth System Model version 2 (NorESM2)—with a spatially disaggregated Integrated Assessment Model (IAM), the Disaggregated Integrated Assessment Model (DIAM). The resulting modeling tool, NorESM2-DIAM, incorporates state-of-the-art climate and weather dynamics, allows economic impacts to depend on the full distribution of weather outcomes, and captures realistic spatial heterogeneity. To our knowledge, it is the first framework to fully couple an ESM with a high-resolution IAM. The primary contribution of this paper is to develop and implement the methodology that

enables this coupling. We demonstrate the utility of NorESM2–DIAM through a baseline simulation. The results show that the economic impacts of global warming vary dramatically across space and that internal climate variability generates substantial volatility in regional GDP, highlighting the importance of high-resolution economic impact assessments. Although the baseline simulation focuses on regional temperature, the framework can be easily extended to incorporate additional variables such as precipitation and extreme events. It can also be applied to study a wide range of climate policies. NorESM2-DIAM represents

an important step towards improving the understanding of economic impacts of climate change and can ultimately become an important source of information for decision-makers.

*Copyright statement.* TEXT

## 1 Introduction

Integrated Assessment Models (IAMs) and Earth System Models (ESMs) are two key tools for investigating the complex

challenges posed by climate change. IAMs focus on the human aspect of climate change; ESMs focus on the natural climate system. With their differing foci and application areas, they are usually developed independently of each other. However, both





modeling tools depend on information from the other, and cooperation between the ESM and IAM communities is essential to understanding the full picture of climate change (e.g. van Vuuren et al., 2012; Calvin and Bond-Lamberty, 2018; Collins et al., 2015; Keen et al., 2021).

IAMs have played a fundamental role in climate economics since the introduction of the DICE model (Nordhaus, 1991, 1992). These models simulate the economic impacts of climate change, allowing evaluation of the policy consequences and the feasibility of various climate targets, and have produced influential results—most notably, estimates of the social cost of carbon—that continue to inform climate policy (e.g. Weyant, 2017; Harremoës and Turner, 2001; Schneider, 1997; Nordhaus, 2010). They usually consist of three separate elements: a climate system, a carbon cycle, and an economic model (van Vuuren et al.,

2011; Nordhaus, 1992). To maintain computational tractability, however, most IAMs rely on highly simplified representations of the climate system and the carbon cycle (Goodess et al., 2003; van Vuuren et al., 2012). Although practical, these simplifications often suppress both spatial and temporal variability and limit the models' ability to capture the complex dynamics of Earth system processes. This, in turn, limits their ability to accurately estimate the economic impacts of climate change.

One key limitation of many IAMs is their sensitivity to how climate dynamics are simplified. For example, they often cannot

reproduce either the delay between carbon emissions and warming or climate–carbon feedbacks, with important consequences for estimates of the social cost of carbon (Dietz et al., 2021; van Vuuren et al., 2011). Furthermore, most IAMs model very few climate variables—primarily global mean surface temperature—despite growing recognition that economic outcomes are shaped not only by long-term temperature trends but also by short-term, high-impact weather events such as heatwaves, storms, floods and droughts (Callahan and Mankin, 2022; Frame et al., 2020). These events reflect local deviations from climatological

averages and often cause severe economic damages. Finally, IAMs typically operate on coarse spatial scales, which restricts their capacity to resolve regional variations in both physical climate responses and economic vulnerabilities (e.g. Dell et al., 2014; Hsiang et al., 2017). Even when regional detail is included, it is often through stylized approaches that link local climate directly to global averages, missing important regional dynamics.

In contrast, ESMs simulate the physical climate system based on fundamental geophysical principles. They include interac-

tive components of the atmosphere, land, ocean, and cryosphere, as well as representation of aerosols, atmospheric chemistry, and the carbon cycle (Taylor et al., 2012; van Vuuren et al., 2011). With their many components, ESMs produce high-resolution outputs across a wide range of climate variables, including realizations of weather and extremes. However, ESMs lack human components and rely on externally prescribed pathways for emissions and land use (e.g. Moss et al., 2010; O'Neill et al., 2017; Riahi et al., 2017), thus excluding any climate–economy interactions.

Motivated by these shortcomings, we present a novel framework that couples the Norwegian Earth System Model version 2 (NorESM2) with a spatially disaggregated, dynamic model of the global economy. We refer to this framework as NorESM2–DIAM, where DIAM stands for Disaggregated Integrated Assessment Model and builds on the high-resolution climate–economy model developed in Krusell and Smith (2022). The coupling replaces stylized climate dynamics, carbon cycle approximations, and simplified regional representations with physically-based, high-resolution outputs from NorESM2.

It also allows economic outcomes to depend on a wide variety of climate and weather variables, including extreme events. This framework thus delivers a new model that can be used to investigate the economic impacts of climate change both globally and





regionally, incorporating both climate-economy feedbacks and internal variability. Finally, NorESM2–DIAM is a cost-benefit IAM capable of evaluating the welfare effects across time and space of a wide range of scenarios for climate policy—from laissez-faire to optimal carbon taxation.

However, the primary goal of this paper is to demonstrate, using a prototype version of NorESM2–DIAM, how to tackle two key methodological challenges in coupling an ESM with a high-resolution, dynamic economic model. First, the two models operate on vastly different time scales. Second, the economic model incorporates forward-looking behavior: the decisions of agents (consumers and firms) depend on their expectations about the future behavior of the climate, which is itself influenced by those very decisions. Achieving consistency between agents' expectations and the climate trajectory thus requires solving

for an interdependent equilibrium.

Successfully addressing these challenges lays the groundwork for using NorESM2–DIAM as a platform to explore the spatial and temporal dimensions of climate–economy interactions, and to assess climate policy with a degree of geophysical and economic realism that is rare in existing IAMs. This platform contributes to a small but growing literature using dynamic structural models to study the spatial effects of climate change (see, for example,  Brock et al., 2014; Desmet and Rossi-

Hansberg, 2015; Fried, 2022; Krusell and Smith, 2022; Rudik et al., 2021; Bilal and Rossi-Hansberg, 2023; Cruz and Rossi-Hansberg, 2024; Kubler, 2023; Kotlikoff et al., 2024).

The rest of the paper is organized as follows: Section 2 describes the two components of our new framework, NorESM2–DIAM. Section 3 explains in detail the interactive coupling between the two components of NorESM2–DIAM. Section 4 discusses the calibration of the model parameters. Section 5 presents quantitative results. Section 6 discusses limitations and

directions for further research using this new platform and, finally, Section 7 offers some concluding remarks.

## 2   Model Description

This section describes the structure and functioning of NorESM2–DIAM, a coupled model that integrates climate and economic dynamics. Section 2.1 begins with an overview of the overall framework. Section 2.2 then outlines the economic component, DIAM, and its coupling with NorESM2. Section 2.3 introduces a standalone version of DIAM used to compute equilibrium

behavior in the fully coupled system. Finally, Section 2.4 provides a description of the Norwegian Earth System Model version 2 (NorESM2).

### 2.1   Overview

The two components of NorESM2–DIAM are coupled via a continuous, bidirectional flow of information. In each time period, economic agents in DIAM—households and firms—make decisions about energy use and other economic variables, taking into

account how local climate and weather conditions affect productivity. The resulting carbon emissions are passed to NorESM2, which simulates the climate response. These outcomes then feed back into DIAM, influencing future economic decisions and creating a dynamic feedback loop between the climate and the economy.





Agents in DIAM solve forward-looking dynamic optimization problems, with decisions shaped by expectations about the future evolution of climate and weather. To make these problems computationally tractable, we employ a standalone version of DIAM with simplified climate dynamics, following standard practice in integrated assessment modeling. In the standalone model, agents' decision-making relies on statistical temperature forecasts that approximate the behavior of NorESM2. Agents' energy use decisions and consequent emissions then serve as input to NorESM2 itself when it updates climate and weather variables.

An economic equilibrium in NorESM2–DIAM is defined by a fixed-point condition: agents' expectations about future climate and weather must align with the outcomes that emerge from their own economic behavior—particularly their choices regarding energy use and hence emissions. The standalone version of DIAM is instrumental in computing this equilibrium.

### 2.2 The economic model (DIAM)

This section describes DIAM, a spatially disaggregated model of the global economy that interacts with climate and weather.

#### 2.2.1 Space and Time

DIAM divides the globe into $M$ regions, each corresponding to a $1° \times 1°$ land cell with observed economic activity in 1990. Cells spanning multiple countries are subdivided along national borders, yielding approximately 19,000 distinct regions. Time proceeds in discrete annual periods and begins in period 0.

#### 2.2.2 Production technology

Each region $i$ contains two production sectors: one for final goods and one for energy. Energy is used as an input in both sectors. In each year $t$, the final goods sector produces output $y_{it}$ using three factors of production (physical capital, labor, and energy):

$$y_{it} = F(k_{it}^y, L_{it}^y, x_{it}^y),$$

where $k_{it}^y$, $L_{it}^y$, and $x_{it}^y$ are the amounts of physical capital, labor, and energy, respectively, used in the final-goods sector and $F$ is the production function mapping inputs to output.

The energy sector uses the same types of inputs to produce energy $x_{it}$:

$$x_{it} = \zeta^{-1} F(k_{it}^x, L_{it}^x, x_{it}^x),$$

where $k_{it}^x$, $L_{it}^x$, and $x_{it}^x$ are the amounts of physical capital, labor, and energy, respectively, used in the energy sector and $\zeta$ denotes the relative productivity of the final-goods sector compared to the energy sector. To simplify the structure, we use the same production function in each sector, up to the relative productivity shifter $\zeta$, but this assumption is not essential to the analysis undertaken here.





When inputs are efficiently allocated across sectors, net final-goods output (i.e., gross domestic product, or GDP) is given by:

$$y_{it} = F(k_{it}, L_{it}, x_{it}) - \zeta x_{it},$$

where $k_{it} = k_{it}^y + k_{it}^x$, $L_{it} = L_{it}^y + L_{it}^x$, and $x_{it} = x_{it}^y + x_{it}^x$ denote the total amounts of capital, labor, and energy used in the region. This allocation can be decentralized as a competitive equilibrium in which firms in both sectors choose their factors of production to maximize profits, taking factor prices as given, and prices adjust to clear factor markets.

Regional population $N_{it}$ evolves exogenously over time. Labor productivity consists of two components: (i) an exogenous factor $A_{it}$ capturing socioeconomic and technological influences outside the scope of the model, and (ii) a climate-sensitive factor that depends on local climate. In the current prototype of NorESM2–DIAM, the latter depends solely on average annual temperature $T_{it}$. Effective labor input is therefore defined as:

$$L_{it} = N_{it} A_{it} D(T_{it}),$$

where $D(\cdot)$ is a "damage function" capturing the effect of temperature on labor productivity. As described in detail in Section 4, $D$ has an inverse $U$-shape with a maximum at $T^*$ so that productivity declines monotonically as regional temperature deviates from $T^*$.

Each period, agents allocate the output of final goods between consumption $c_{it}$ and investment $\iota_{it}$, subject to a budget constraint:

$$y_{it} = c_{it} + \iota_{it}.$$

Capital depreciates at a constant rate $\delta$, with investment replenishing the capital stock:

$$k_{i,t+1} = (1 - \delta)k_{it} + \iota_{it}.$$

### 2.2.3 Carbon emissions

Energy use is measured in gigatons of oil equivalent (Gtoe), where one Gtoe corresponds to $3.97 \times 10^{16}$ BTUs (British Thermal Units). One unit of energy use releases $\psi \phi_t$ gigatons of carbon emissions (GtC) into the atmosphere in period $t$. The fraction $\phi_t$ is normalized to 1 in period 0, so that $\phi_t$ measures the "dirtiness" of energy use relative to period 0. From period $t_g$ onward, all energy is assumed to be fully green, implying $\phi_t = 0$ for all $t \geq t_g$. The full trajectory of $\phi_t$ is described in Section 4.

Total global emissions in period $t$, denoted $E_t$, are the sum of regional emissions:

$$E_t = \sum_{i=1}^{M} e_{it}, \quad \text{where } e_{it} = \psi \phi_t x_{it}.$$

### 2.2.4 Markets

This version of DIAM excludes international capital markets, so capital is immobile across regions. As noted by Krusell and Smith (2022), this simplification has only minor effects on the dynamics examined here. Consequently, regions interact





solely through the climate system, whose evolution is determined by their collective energy-use choices and associated carbon emissions. Regions can adapt to climate change by adjusting their capital accumulation and energy-use decisions accordingly.

### 2.2.5   Preferences

Agents in each region seek to maximize welfare, defined here as the expected discounted utility of consumption:

$$\mathbb{E}_0 \sum_{t=0}^{\infty} \beta^t N_{it} U(c_{it}),$$

where $\beta \in [0,1)$ is the discount factor, $U(c_{it})$ is per capita utility from consumption, and $\mathbb{E}_0$ denotes expectations formed at time 0. They do so by choosing paths for consumption, investment, and energy use subject to the production and capital accumulation constraints outlined in Section 2.2.2. This is a dynamic optimization problem in which agents' decisions hinge on expectations about future labor productivity, which is itself influenced by climate conditions. Agents can therefore adapt to changes in regional climate by shifting their patterns of consumption, investment, and energy use over time.

### 2.2.6   Climate feedback


Conventional IAMs link the economy and climate through a simplified climate system and carbon cycle: global emissions feed into this reduced-form climate representation, which then produces projections of climate variables—typically global and regional mean temperatures. These climate variables, in turn, affect economic productivity. While computationally convenient, this leaves out many of the complex, region-specific processes that drive climate change and its impacts.

Our approach replaces this simplification with a direct coupling between regional emissions and NorESM2, a state-of-the-art Earth System Model described in Section 2.4. NorESM2 resolves the physical climate system in much greater detail, simulating geophysical responses—including temperatures—at high spatial and temporal resolution. This enables the economic model to respond to a climate signal that captures fine-scale processes, regional heterogeneity, and nonlinear interactions often omitted in IAMs.

Realizing this coupling requires overcoming two key challenges. First, DIAM operates on annual decision intervals, whereas NorESM2 advances in much finer time steps—hourly or sub-hourly. Within each DIAM year, NorESM2 must run many high-frequency computations before returning updated climate and weather inputs. Section 3 describes how this temporal mismatch is reconciled.

Second, NorESM2's high-dimensional, nonlinear dynamics make it computationally impossible to embed directly into
DIAM's regional optimization problem. To address this difficulty, the next section develops a standalone version of DIAM with a simplified climate representation calibrated to remain broadly consistent with NorESM2. This allows agents to form approximate yet reliable climate forecasts, keeping their decisions close to the true optimum while making the coupled system computationally feasible.





### 2.3 The standalone version of DIAM

This section describes a simplified version of DIAM in which agents forecast regional temperatures using a reduced-form climate model. This standalone model facilitates solving the regional optimization problems and enables computing an economic equilibrium—defined as a fixed point where agents' forecasts align with realized outcomes. Computing this fixed point offers a tractable approach for approximating equilibrium in the full coupled model.

Section 2.3.1 introduces the statistical approach used to forecast regional temperatures, Section 2.3.2 outlines the optimiza-
tion problem faced by agents, and Section 2.3.3 explains how equilibrium is computed in the standalone setting.

#### 2.3.1 Statistical temperature forecasting

In the standalone DIAM, agents forecast regional temperatures—key determinants of productivity—using a low-dimensional statistical approach. This forecast approximates NorESM2's geophysical dynamics and can be calibrated using both historical and future scenario simulations of NorESM2.

The model has two components. The first relates the expected regional temperature in region $i$ at time $t$ to cumulative global carbon emissions since the pre-industrial era:

$$\overline{T}_{it} = \overline{T}_i + \gamma_{i1}S_t + \gamma_{i2}S_t^2, \tag{1}$$

where $\overline{T}_i$ is the pre-industrial temperature in region $i$ and $S_t$ denotes cumulative global emissions up to the beginning of period $t$. The functional form and the parameters $\gamma_{i1}$ and $\gamma_{i2}$ are estimated from data, as described in Section 4.

The second component captures deviations from the expected temperature due to internal climate variability. While these fluctuations follow nonlinear, non-stochastic laws of motion in NorESM2, they are modeled stochastically here as an AR(1) process:

$$z_{i,t+1} = \rho_i z_{it} + \epsilon_{i,t+1}, \tag{2}$$

where $\{\epsilon_{it}\}$ is a sequence of independent, normally distributed shocks with mean zero and standard deviation $\sigma_i$. The realized
temperature is then:

$$T_{it} = \overline{T}_{it} + z_{it}.$$

The parameters $\rho_i$ and $\sigma_i$ are also estimated using data from NorESM2 simulations (see Section 4). We assume that these parameters remain constant as the climate warms, and leave the exploration of potential deviations from this assumption to future work.

#### 2.3.2 Dynamic optimization

To make forward-looking decisions, agents form expectations about future temperatures using the statistical model introduced in Section 2.3.1. Because a single region's emissions have a negligible effect on global totals, agents treat the sequence of



cumulative global emissions $\mathbb{S} \equiv \{S_t\}_{t=0}^{\infty}$ as exogenous when making decisions. Furthermore, since annual fluctuations in global emissions are small relative to their overall level, agents assume that this sequence is deterministic. These assumptions enable them to forecast the expected component of regional temperature.

Given physical capital and effective labor, agents choose energy use to maximize net output of final goods. This static optimization yields a decision rule for energy use:

$$h_{it}^x(k_{it}, z_{it}) = \arg\max_{x_{it}} [F(k_{it}, L_{it}, x_{it}) - \zeta x_{it}], \tag{3}$$

where

$$L_{it} = N_{it} A_{it} D(\overline{T}_{it} + z_{it}). \tag{}$$

The decision rule for optimal energy use is time-varying because effective labor $L_{it}$ depends on population $N_{it}$, exogenous productivity $A_{it}$, and expected temperature $\overline{T}_{it}$—all of which follow deterministic paths that agents take as given when solving their optimization problems.

The investment decision is dynamic and is characterized recursively by the Bellman equation:

$$v_{it}(\omega_{it}, z_{it}) = \max_{k_{i,t+1}} [N_{it} U(\omega_{it} - k_{i,t+1}) + \mathbb{E}_t(v_{i,t+1}(\omega_{i,t+1}, z_{i,t+1}))],$$

where $v_{it}(\omega_{it}, z_{it})$ is the value function, representing the optimal expected utility from period $t$ onward, given current wealth $\omega_{it}$ and the current temperature deviation $z_{it}$, which serves as a sufficient statistic for computing forecasts of future temperature deviations. Regional wealth $\omega_{it}$ in any period $t$ is defined as:

$$\omega_{it} = F(k_{it}, L_{it}, x_{it}) - \zeta x_{it} + (1 - \delta)k_{it}, \tag{4}$$

with energy use $x_{it}$ chosen optimally according to the static decision rule.

The expectations operator $\mathbb{E}_t$ in the Bellman equation denotes integration of $v_{i,t+1}(\omega_{i,t+1}, z_{i,t+1})$ over the conditional distribution of $z_{i,t+1}$ given $z_{it}$. According to the statistical temperature forecast in Section 2.3.1, this distribution is normal with mean $\rho z_{it}$ and standard deviation $\sigma_i$. Note that the value function in period $t + 1$ depends both directly and indirectly on $z_{i,t+1}$, since future wealth $\omega_{i,t+1}$ also varies with $z_{i,t+1}$.

Solving the Bellman equation yields a decision rule for investment:

$$k_{i,t+1} = h_{it}^k(\omega_{it}, z_{it}), \tag{5}$$

This decision rule is time-varying, as it depends on the entire forward-looking path—from period $t$ onward—of population, exogenous productivity, and expected temperature, all of which evolve deterministically. Each region faces a distinct optimization problem due to differences in these trajectories and in the region-specific parameters that govern temperature forecasts: $\overline{T}_i, \gamma_{i1}, \gamma_{i2}, \rho_i,$ and $\sigma_i$. We solve each region's dynamic programming problem numerically using the endogenous grid method (see Appendix A for details). Solving the approximately 19,000 dynamic programs in parallel is straightforward and takes about five minutes in Julia using 80 cores on Yale University's High Performance Cluster.



### 2.3.3 Equilibrium

This section examines equilibrium in the standalone model under the assumption that the statistical temperature forecasting
approach in Section 2.3.1 accurately describes the actual evolution of regional temperatures. Under this assumption, the standalone model is fully self-contained and requires no interaction with NorESM2 in any period. We further assume that, when
simulating the standalone model, there is no internal variability: all realized temperature deviations $z_{it}$ are set to zero, so that
realized regional temperatures exactly match their expected values in every period. Under these conditions, a perfect-foresight
equilibrium is defined as a fixed point in the sequence $\mathbb{S}$ of cumulative global emissions: agents take $\mathbb{S}$ as given when solving
their optimization problems, and their optimal decisions reproduce the same sequence $\mathbb{S}$.

To compute this equilibrium, we begin with an initial guess for the global emissions sequence $\{E_t^{(0)}\}_{t=0}^{T}$, truncating the
infinite horizon at $T > t_g$ so that $E_t = 0$ when $t_g \leq t \leq T$. Using this sequence we calculate the corresponding cumulative
emissions sequence $\mathbb{S}^{(0)}$ via $S_{t+1}^{(0)} = S_t^{(0)} + E_t^{(0)}$ with $S_0^{(0)}$ predetermined. Given $\mathbb{S}^{(0)}$, each region's decision rules are computed
by solving its dynamic programming problem backward from $t = T$ to $t = 0$ as described in Appendix A. We then simulate the
global economy forward in time from the initial regional capital stocks, imposing $z_{it} = 0$ for all $i$ and $t$. Energy use and capital
accumulation follow $x_{it} = h_{it}^x(k_{it}, 0)$ and $k_{i,t+1} = h_{it}^k(\omega_{it}, 0)$, where wealth $\omega_{it}$ is given by eq. 4 with $L_{it} = N_{it}A_{it}D(\overline{T}_{it})$.

The forward simulation produces a new global emissions sequence $\{E_t^{(1)}\}_{t=0}^{T}$ and corresponding cumulative emissions $\mathbb{S}^{(1)}$.
If $\mathbb{S}^{(1)}$ is within the chosen convergence tolerance of $\mathbb{S}^{(0)}$, we take $\mathbb{S}^{(1)}$ as the equilibrium sequence $\mathbb{S}^*$. Otherwise, we update
the guess by replacing $\mathbb{S}^{(0)}$ with $\mathbb{S}^{(1)}$ and repeat the backward–forward iteration until convergence. In practice, this algorithm
converges after a small number of iterations (typically five or fewer). The resulting $\mathbb{S}^*$ serves as a candidate for the equilibrium
path of cumulative emissions in the fully coupled NorESM2–DIAM model.

### 2.4 The Norwegian Earth System Model version 2 (NorESM2)

Earth System Models (ESMs) are state-of-the-art computational models designed to simulate and understand the dynamics
of the climate system. They divide the atmosphere, ocean, and land surface into a three-dimensional grid, within which they
numerically solve equations that describe key physical, chemical, and biological processes. These calculations are typically
performed at hourly or even sub-hourly time steps, generating high-resolution output across time and space for a wide range of
climate variables, including temperature, precipitation, wind, and carbon stocks. Processes that cannot be resolved directly—
either because they occur at scales smaller than the grid or are too complex to model in full detail—are represented using
simplified representations known as parameterizations.

NorESM2 is an ESM with coupled atmosphere, ocean, land, river transport, and sea ice (for full description see Seland et al.,
2020), which contributed to the 6th phase of the Coupled Model Intercomparison Project (CMIP6; Eyring et al., 2015). It is
largely based on the Community Earth System Model version 2 (CESM2; Danabasoglu et al., 2020) with two main differences:

1. The atmospheric component, the Community Atmosphere Model (CAM6) is replaced by CAM6-Nor, which includes
   several modifications from CAM6: A different aerosol module OsloAero6 (Kirkevåg et al., 2018), specific modifications





and tunings of the atmosphere component (Toniazzo et al., 2020), as well as an updated parameterisation of turbulent
        air-sea fluxes (developers group, 2021).

    2. A different ocean model with isopycnic coordinates: The Bergen Layered Ocean Model (BLOM; developers group,
       2021), which is coupled to the Hamburg Ocean Carbon Cycle Model (Tjiputra et al., 2020).

        We use the NorESM2-LME configuration, which has an active carbon cycle enabled. This version has 1° ocean and sea ice
resolution, 2° atmosphere and land resolution, 32 vertical layers in the atmosphere, a 30-minute timestep for atmosphere, land,
        and sea ice, and a 1-hour timestep for the ocean (Seland et al., 2020). When the carbon cycle is active, the model calculates
        greenhouse gas concentrations from spatial emissions, otherwise, the greenhouse gas concentrations must be prescribed.

        The climate of NorESM2 has been assessed through various experiments (see Eyring et al., 2015, for details of the bench-
        mark simulations). Its historical simulations follow the observations relatively well, although NorESM2 has a weaker warming
between 1930 and 1970 than the observations (Seland et al., 2020). Simulations with a 1% increase in $CO_2$ each year until
        doubling, and with an abrupt quadrupling of $CO_2$ were performed to calculate the transient climate response (TCR; the tem-
        perature change at doubling of $CO_2$ without climate stabilization) and an approximation of the equilibrium climate sensitivity
        (ECS; temperature change at doubling of $CO_2$ after the climate has stabilised), respectably (Eyring et al., 2015). In NorESM2,
        TCR is 1.48K and ECS is 2.54K (Seland et al., 2020), which is within the likely ranges estimated by the IPCC (Forster et al.,
2021)—although at the lower end.

### 2.4.1   The carbon cycle

The carbon cycle describes the exchange and storage of carbon between the atmosphere, ocean, land surface, and lithosphere
through physical, chemical, and biological processes. The carbon cycle includes several feedbacks with other components of
the climate system: Changes in climate due to increased carbon in the atmosphere (e.g., increasing temperature or changing pre-
cipitation) may change the rate of carbon transfer (e.g., photosynthesis) or carbon storage capacity (e.g., forest or permafrost),
thus either amplifying or reducing the initial increase.

        In the carbon cycle of NorESM2, anthropogenic carbon emissions from fossil fuel combustion and changes in land use are
prescribed. The carbon cycle is represented mainly through the land model (CLM5; Lawrence et al., 2019) and the biogeochem-
ical component (iHAMOCC) coupled with the ocean component (BLOM) (Tjiputra et al., 2020). The land model simulates
the uptake, storage, and release of carbon from vegetation and soil through photosynthesis, respiration, and decomposition,
as well as nitrogen cycling and disturbances such as forest fires (Lawrence et al., 2019). The biogeochemical component cal-
culates the partial pressure of $CO_2$ based on temperature, salinity, dissolved inorganic carbon, and alkalinity, and uses it to
calculate the air-sea $CO_2$ fluxes (Tjiputra et al., 2020). An ecosystem module simulates phytoplankton and zooplankton with
limiting nutrients (nitrate, phosphate, and dissolved iron), dissolved organic carbon, and particulate matter (developers group,
2021). Furthermore, there are vertical fluxes of organic and inorganic carbon, and the former is remineralized at a given rate
(Schwinger et al., 2016). Finally, a sediment module collects the non-remineralized particle matter (developers group, 2021).





### 2.4.2 Preparing NorESM2 for coupling

Before NorESM2 was ready for use, it was initialized and run to a steady state—a process known as the spin-up. This was carried out by the NorESM2 developers' group (developers group, 2021; Seland et al., 2020) in parallel with the final calibrations

of the model: They initialized NorESM2—without an active carbon cycle—with a combination of previous model simulations and observational estimates for the pre-industrial climate (i.e., year 1850). Human emissions of greenhouse gases and aerosols were set to zero or close to zero. The model was then run for more than 1000 years to allow the climate to reach a steady state. During the spin-up, some additional model parameters were tuned to reach this steady state. Finally, the carbon cycle was turned on and the model ran another for 100 years.

For the prototype NorESM2–DIAM coupling, only $CO_2$ emissions are included. The model is initialized from 1850 pre-industrial conditions, with all non-$CO_2$ anthropogenic forcings fixed at 1850 levels. $CO_2$ emissions follow the CMIP6 historical dataset (Eyring et al., 2015) until 1990, after which they are endogenously determined by the energy-use decisions of agents. An extension to NorESM2-DIAM which links economic activity to the emission of other climate forcing agents than $CO_2$ will be the focus of future research.

No changes to the model code were necessary for the purpose of the coupling.

## 3 Coupling

This section describes how the two components of NorESM2–DIAM are coupled. The procedure has three stages: (1) compute the equilibrium of the standalone DIAM, which serves as a candidate equilibrium for the coupled model; (2) run a dynamic simulation of the fully coupled system, exchanging information between the two components at each annual time step; and (3)

assess the accuracy of the candidate equilibrium.

In the first stage, we solve for the equilibrium path of cumulative emissions, $\mathbb{S}^*$, in the standalone DIAM without internal variability, as described in Section 2.3.3. Doing so yields time-varying regional decision rules for optimal investment and energy use, given $\mathbb{S}^*$. These rules rely on the temperature forecasting approach described in Section 2.3.1; however, in the fully coupled NorESM2–DIAM, climate and weather outcomes are generated by the complete NorESM2 dynamics.

In the second stage, we simulate the coupled global economy–climate system on a year-by-year basis using the decision rules computed in the first stage. A key challenge is a temporal mismatch: NorESM2 operates on hourly or sub-hourly time steps, while DIAM operates annually. Consequently, economic decisions are fixed once per year and cannot respond to intra-annual weather fluctuations.

To address this temporal mismatch, we assume that regional energy use in year $t$ equals its conditional expectation, formed

in year $t-1$ based on the year-$t$ capital stock $k_{it}$ (set the previous year) and the previous year's temperature deviation $z_{i,t-1}$ from its expected value in year $t-1$ as specified in eq. (1). Using the decision rule for energy use, this conditional expectation, to first order, is:

$$\bar{x}_{it} = h_{it}^x\big(k_{it}, E_{t-1}(z_{it})\big), \quad E_{t-1}(z_{it}) = \rho_i\, z_{i,t-1}.$$





NorESM2 uses the resulting regional emissions $\phi_{it}\bar{x}_{it}$—distributed evenly across the year—to advance the climate by one

year, producing high-frequency weather data, including temperature, for each sub-period. The realized annual average temperature in region $i$ is denoted $T_{it}$. Actual energy use is then

$$x_{it}^z = h_{it}^x(k_{it}, z_{it}),$$

where $z_{it}$ is the deviation of $T_{it}$ from its expected value, $\overline{T}_{it}$, in year $t$.

Reconciling the temporal mismatch in this way introduces a small gap between actual and expected emissions, given by

$x_{it}^z - \bar{x}_{it}$. This gap averages to zero over time, and each year's gap is added to the next year's expected emissions for consistency.

Regional wealth evolves each year according to

$$\omega_{it} = F\big(k_{it}, N_{it}A_{it}D(T_{it}), x_{it}^z\big) - px_{it}^z + (1-\delta)k_{it}. \tag{6}$$

Once the state variables $\omega_{it}$ and $z_{it}$ have been updated, the model advances to the next annual cycle. The capital stock for year $t+1$ is then determined by the investment rule in eq. (5), completing the sequence of yearly decisions.

Appendix A contains additional details on how we execute the coupled simulation.

In addition to the temporal mismatch, there is also a spatial mismatch: DIAM operates at a $1°$ resolution, while NorESM2 uses $2°$ resolution for its atmosphere and land components. We reconcile these grids using linear interpolation.

In the third and final stage, we evaluate the accuracy of the candidate equilibrium by assessing whether the behavior of the fully coupled simulation aligns with the temperature forecasts that agents rely on to make optimal decisions. The main

task is to compare the cumulative emissions path from the coupled run, $\mathbb{S}^c$, with the fixed-point path $\mathbb{S}^*$ from the standalone DIAM. Internal variability—absent in the calculation of $\mathbb{S}^*$—introduces persistent weather-driven fluctuations in the coupled run, producing year-to-year variation in global emissions and hence in $\mathbb{S}^c$. These deviations, however, are small and agents would gain little from incorporating them into the temperature forecasts guiding their decisions. Apart from these minor discrepancies, $\mathbb{S}^c$ and $\mathbb{S}^*$ track each other closely, as we discuss in detail in Section 5. Consequently, we find that there is no need

to refine the candidate equilibrium through additional iterations between successive full-model simulations. This conclusion has important practical significance for our methodology: running NorESM2–DIAM is computationally demanding—each annual cycle requires about an hour on a supercomputer—whereas computing the candidate equilibrium in the standalone model typically takes less than an hour in total.

## 4 Calibration

This section details the calibration of parameters in DIAM, including the temperature forecasting approach used in the standalone version. It also outlines the choice of initial regional capital stocks and the construction of projected trajectories for regional population and the exogenous component of regional productivity. The base year in the calibration is 1990, the earliest date for which we have regional data on output and population.





## 4.1 Production technology

The production function $F(k, L, x)$ is Cobb-Douglas and exhibits constant returns to scale:

$$F(k, L, x) = \left(k^\alpha L^{1-\alpha}\right)^\theta x^{1-\theta}, \tag{7}$$

where $\alpha$, capital's share of income (or GDP), is set to 0.36.

The productivity of the final-good sectors relative to the energy sector, $\zeta$, equals the equilibrium price, $p$, of a unit of energy. Let $s_{it}$ denote the share of energy production in region $i$ in year $t$, measured as a fraction of regional GDP:

$$s_{it} = \frac{px_{it}}{y_{it}}.$$

In equilibrium, each region chooses energy to solve the optimization problem in eq. (3). The first-order condition for this problem sets the marginal product of energy equal to $p$:

$$F_x(k_{it}, L_{it}, x_{it}) = p. \tag{8}$$

This condition implies that the energy share is constant:

$$s_{it} = \frac{\theta}{1 - \theta} \equiv s.$$

Following Krusell and Smith (2022), we set $\theta = 0.058$, so that $s$ equals the observed global energy share of GDP in 1990.

The price $p$ is pinned down by matching global data from 1990:

$$p = \frac{sY_{i0}}{X_{i0}},$$

where $Y_{i0}$ is global GDP in 1990 (\$36.1 trillion in 1990 according to the G-Econ database discussed in further detail in Section 4.3) and $X_{i0}$ is global energy use in 1990 (10.3 Gtoes as reported in Krusell and Smith (2022)). This gives a calibrated value of $p = 0.203$.

The annual growth rate, $g$, of the exogenous component of labor productivity, $A_{it}$, is set to 1.5% across all years and regions. Allowing for variation across regions and years would be straightforward, but we leave this for future work. Section 4.3 explains how the initial regional levels $A_{i0}$ are determined.

The parameter $\psi$ measuring gigatons of carbon emissions per Gtoe in 1990 is set to 0.586 to match global emissions of 6.03 GtC in 1990. Finally, the annual rate of depreciation of the capital stock, $\delta$, is set to 0.06.

## 4.2 Preferences

The period utility function $U(c) = \log(c)$. Along a balanced growth path (after the transition to clean energy is complete), the equilibrium interest rate $r^* \equiv \beta^{-1}(1 + g) - 1$. We set $\beta = 0.985$, so that $r^* = 0.03$, or 3% per year.





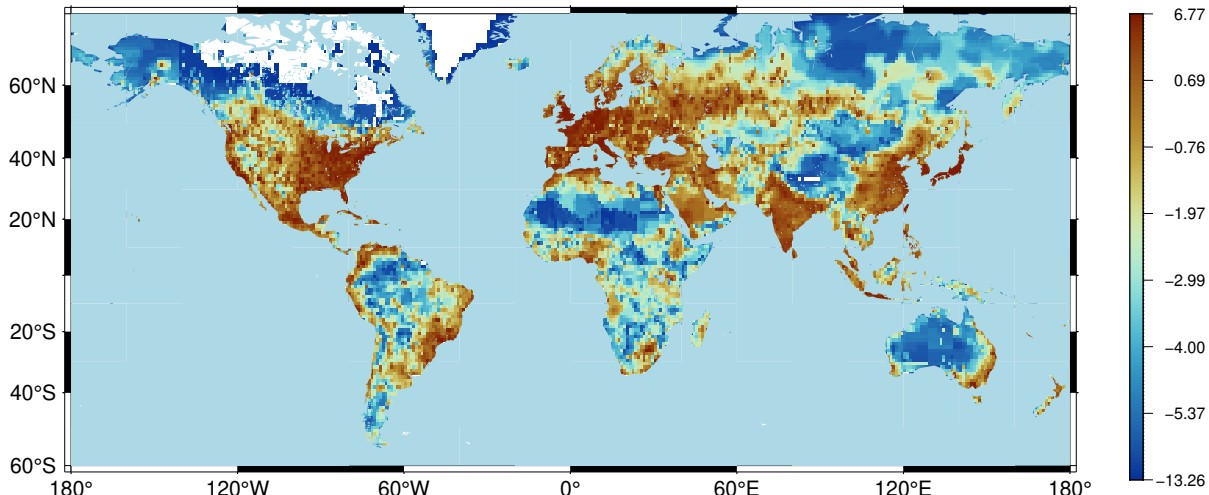

**Figure 1.** Logarithm of regional GDP in 1990.

## 4.3  Initial regional capital stocks and productivity

We use version 4.0 of the G-Econ database (Nordhaus et al., 2006) to obtain regional GDP, $y_{i0}$, and population, $N_{i0}$, in 1990.
Regions with very small populations are excluded, leaving 19,240 regions in total. Figure 1 displays the logarithm of regional GDP in 1990, revealing substantial heterogeneity across space.

Let labor productivity in region $i$ in 1990 be $a_{i0} = A_{i0}D(T_{i0})$. To determine values for $a_{i0}$ and physical capital $k_{i0}$ in 1990, we impose two conditions. First, regional GDP in the model must match GDP $y_{i0}$ in the G-Econ database in 1990:

$$F(k_{i0}, L_{i0}, x_{i0}) - px_{i0} = y_{i0},$$

where $L_{i0} = N_{i0}a_{i0}$ is effective labor in 1990 and the optimal energy choice $x_{i0}$ satisfies the first-order condition in eq. (8).

Second, consistent with the evidence in Caselli and Feyrer (2007), we require that the marginal net return to capital be equalized across regions in 1990:

$$F_k(k_{i0}, L_{i0}, x_{i0}) - \delta = r^*,$$

where $F_k$ is the marginal product of capital. We set the common net return to $r^*$, under the assumption that in 1990 the global economy was approximately on a balanced growth path (when the effects of global warming were still small). The initial value of $A_{i0}$ is then equal to $a_{i0}/D(T_{i0})$, using the damage function $D$ specified in Section 4.5.

## 4.4  Population

To construct time paths for regional population from 1990 to 2140 (the time horizon of DIAM), we proceed in five steps.

**Step 1. Historical data (1990–2005).** We begin with the G-Econ database, which provides regional population data for 1990, 1995, 2000, and 2005. Linear interpolation is used to fill in annual values between these benchmark years.



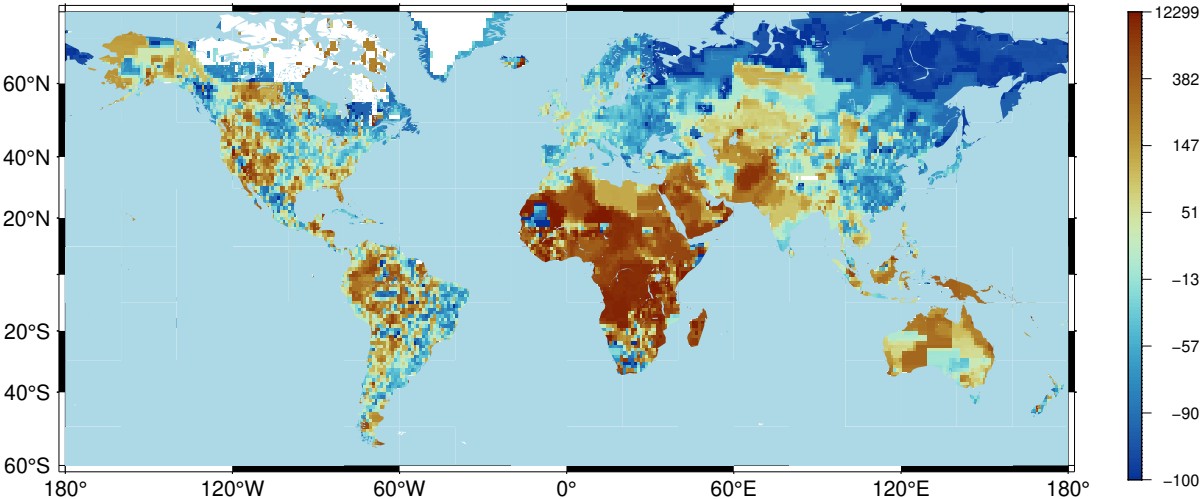

**Figure 2.** Percent change in population from 1900 to 2100. The color bar is not linear; instead each increment in the color bar represents the same number of regions.

**Step 2. Regional shares within countries.** For 1990–2005, we compute each region's share of its country's total population. Because these shares evolve over time, we project them forward to 2100 by assuming that the logarithm of the shares follows a linear trend estimated from the 1990–2005 data. We keep the shares constant after 2100. The key idea is that multiplying these projected shares by country-level population yields regional population after 2005.

**Step 3. Country-level population growth rates.** To construct country-level populations, we first calculate annual growth rates from 2006 to 2100 using data from the 2024 Revision of the United Nations World Population Prospects (United Nations, 2024): historical estimates up to 2024 and projections thereafter. Beyond 2100, we assume that annual growth rates decline linearly to zero by 2140.

**Step 4. Country-level population paths.** Using these growth rates, starting from the G-Econ country-level populations in 2005, we generate annual country-level population paths from 2006 to 2140.

**Step 5. Regional populations.** Finally, we obtain regional populations by multiplying the country-level populations by the projected regional shares.

Figure 2 shows the projected percentage change in regional populations between 1990 and 2100. The results highlight a pronounced demographic shift: sharp population declines in Europe, Russia, and East Asia are counterbalanced by substantial growth in Africa and the Middle East.

## 4.5 Damages and regional temperature

The damage function $D(T_{it})$ captures how labor productivity, measured as a fraction of optimal productivity $A_{it}$ at any point in time, varies with regional temperature $T_{it}$. It has an inverse $U$-shape and is normalized so that its maximum value equals 1



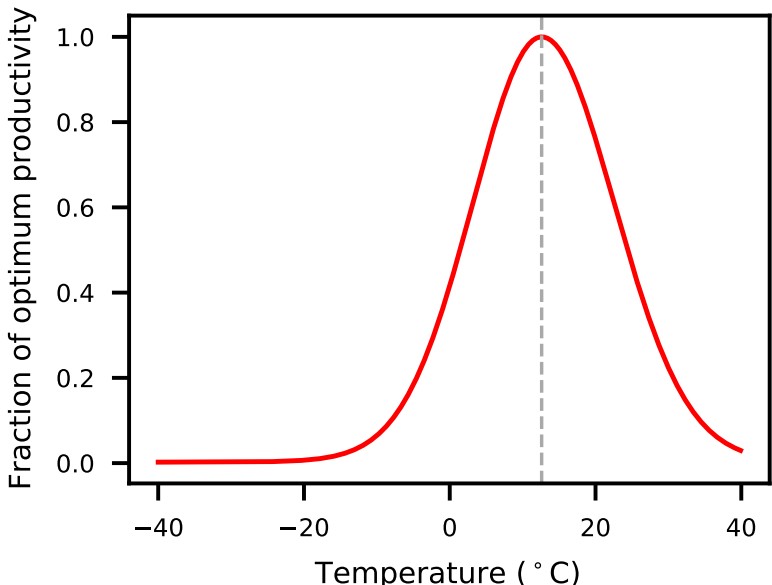

**Figure 3.** The damage function in DIAM as given in eq. (9). The optimum temperature is approximately 12.6°C.

at its peak $T^*$:

$$D(T_{it}) = \begin{cases} \left( (1-d)e^{-\kappa^+ (T_{it}-T^*)^2} + d \right)^{\frac{1}{1-\alpha}} & \text{if } T_{it} \geq T^* \\ \left( (1-d)e^{-\kappa^- (T_{it}-T^*)^2} + d \right)^{\frac{1}{1-\alpha}} & \text{if } T_{it} < T^*, \end{cases} \tag{9}$$

where the parameter $d$ is a lower bound on $D^{1-\alpha}$. The parameters $\kappa^-$ and $\kappa^+$ govern how quickly $D$ declines from its peak to the left and right sides of $T^*$, respectively. Following Bjordal et al. (2022), we set $T^* = 12.61$, $\kappa^- = 0.00328$, $\kappa^+ = 0.00363$, and $d = 0.02$, so the optimum temperature is approximately 12.6°C and the $U$-shape is bounded below by 0.02 and asymmetric, declining more rapidly to the right of the peak than to the left; see Fig. 3.

Figure 4 displays regional productivity using annual temperatures in 1990. There is substantial heterogeneity in productivity across space, reflecting the wide variation in regional temperatures. In addition, comparing to Fig. 1, regions with high GDP in 1990 tend to have productivity near the peak of $D$, while regions with low GDP tend to have lower productivity. The shape of $D$ plays a key role in determining aggregate economic damages from global warming; we discuss these further in Section 5.

### 4.6    The transition to green energy

The economic model assumes that energy use gradually becomes green, represented by the sequence $\{\phi_t\}$. The initial value, $\phi_0$, is normalized to 1, so that the dirtiness of energy use is measured relative to 1990. We assume that $\phi_t = 0$ for $t \geq t_g = 111$, implying that energy use is fully green by the year 2100.



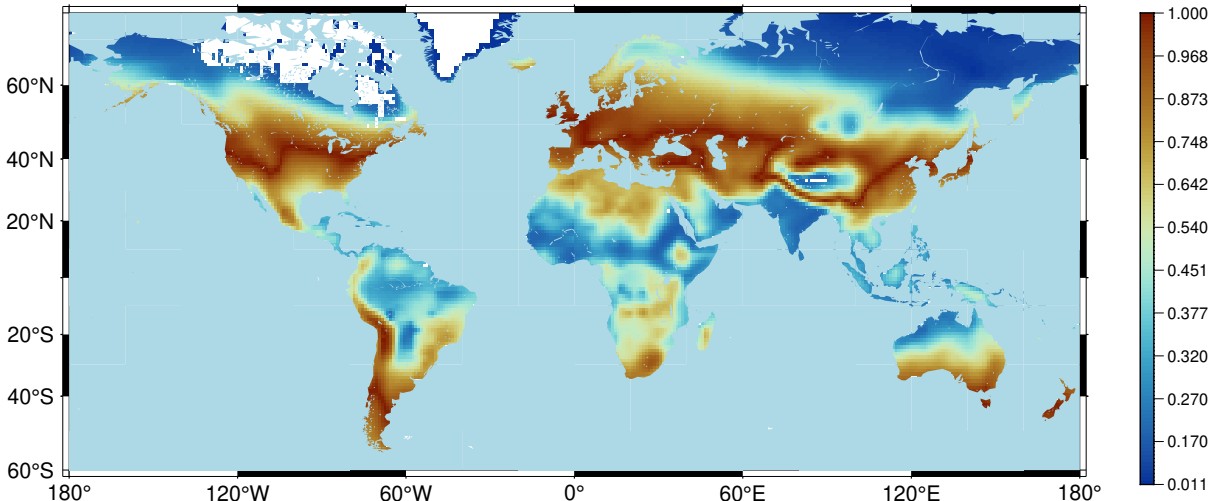

**Figure 4.** Regional productivity in 1990.

To model the transition, we use a logistic function of time:

$$H(t) = \left(1 + \exp\left(\log\left(\frac{0.01}{0.99}\right)\frac{t - n_{0.5}}{n_{0.01} - n_{0.5}}\right)\right)^{-1},$$

with parameters $n_{0.01} = 10$ and $n_{0.5} = 75$. This function is close to 1 when $t = 0$ and declines slowly at first before accelerating, with $H(10) = 0.99$ and $H(75) = 0.5$. For $t < t_g$, we then define

$$\phi_t = \frac{H(t)}{H(0)}.$$

Figure 5 shows $1 - \phi_t$, which we refer to as the greening function. The transition is slow in the early decades: by 2025, only about 5% of energy use is green. This aligns reasonably well with observed data: the share of green energy (renewables and

nuclear) was 11.3% in 1990 and 17.6% in 2024 according to Ritchie and Rosado (2020) so an incremental 5% greening relative to 1990 by 2025 is consistent. After 2025, the pace accelerates, with half of energy use projected to be green by around 2065.

**4.7  Temperature forecasts in the standalone model**

In the standalone version of DIAM, agents form temperature forecasts using a simple statistical approach, described in Section 2.3.1. First, they use cumulative $CO_2$ emissions to project the expected value of regional temperatures, as specified in eq. (1).

Second, they model stochastic deviations from this expected value with an AR(1) process, given in eq. (2).

To ensure consistency with NorESM2—a key requirement when coupling DIAM with the climate model—we estimate the parameters of these two equations using data generated by NorESM2. Specifically, we draw on three NorESM2 simulations with $CO_2$ emissions as the sole forcing. The first simulation begins in 1850, follows historical emissions until 2014 (Eyring et al., 2015), and continues with a future projection of $CO_2$ emissions from SSP3-7.0 for 2015–2100 (van Vuuren et al., 2014;



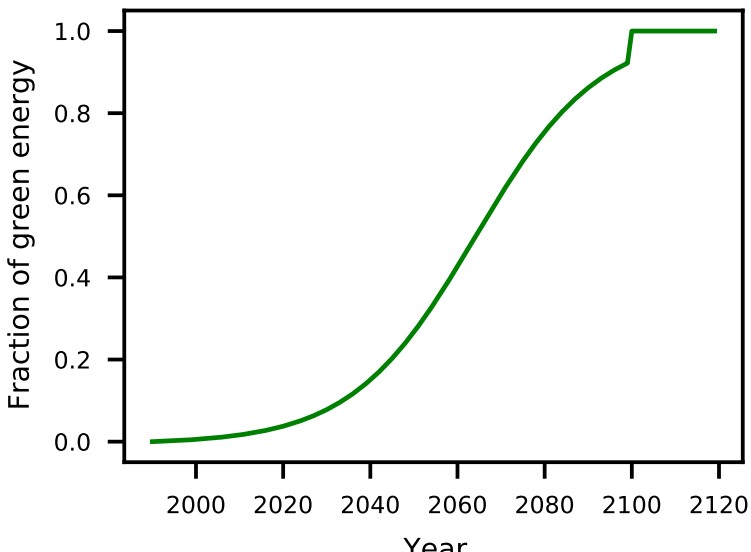

**Figure 5.** The greening function, showing the fraction of energy use that is green.

Kriegler et al., 2014; Riahi et al., 2017). The other two simulations start in 1990 (branching off from the first simulation) and extend to 2100, following lower emissions trajectories derived from standalone DIAM runs. All three simulations are shown in Fig. 6.

The estimation proceeds in two steps. In the first step, we pool the three simulations and regress regional temperature anomalies, $\overline{T}_{it} - \overline{T}_i$, on a quadratic function of cumulative emissions (excluding a constant term), yielding estimates of the

parameters $\gamma_{i1}$ and $\gamma_{i2}$ in eq. (1). In the second step, we compute the residuals from this regression and regress them on their lagged values (again excluding a constant) to estimate $\rho_i$ and $\sigma_i$ in eq. (2).

The relationship between temperature change and cumulative $CO_2$ emissions is referred to as the Transient Climate Response to Cumulative $CO_2$ Emissions (TCRE). It is generally found to be nearly linear, both at the global scale (Matthews et al., 2009; Canadell et al., 2021) and regionally (Leduc et al., 2016). However, as seen in Fig. 6, in NorESM2 the global temperature is

a slightly concave function of cumulative emissions. For that reason, we include a quadratic term in eq. 1, and we find that the estimates of $\gamma_{i2}$, the coefficient on this term, are almost all negative (though small in absolute value). This concavity likely reflects nonlinearities in certain climate feedbacks captured by NorESM2, though exploring these mechanisms lies beyond the scope of this paper.

Finally, Fig. 7 illustrates substantial heterogeneity in regional warming responses: the amount of regional warming associated

with a one-degree increase in global mean temperature (over populated areas only, relative to 1990) varies widely, from less than one degree in much of the Southern Hemipshere to more than one degree—and as high as several degrees—in the northern




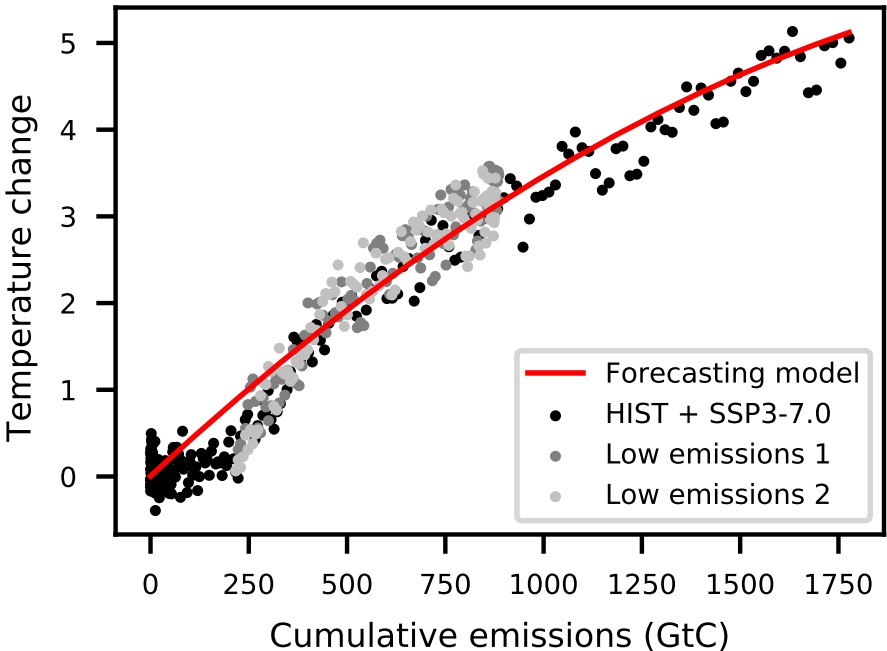

**Figure 6.** NorESM2 global mean temperature change from pre-industrial over land against cumulative emissions since 1850. The red line shows the global mean expected temperature over land obtained by summing eq. (1) across regions. The black dots are from a NorESM2 simulation with historical $CO_2$ emissions followed by the $CO_2$ emissions from SSP3-7.0. The light and dark gray dots are from two NorESM2 simulations from 1990 until 2100 with much lower total emissions at the end of the century.

latitudes. The AR(1) estimates also display heterogeneity: the median estimate of $\rho_i$ is 0.266, with an interquartile range (IQR) of 0.206 to 0.316, while the median estimate of $\sigma_i$ is 0.632, with an IQR of 0.497 to 0.862.

## 5    Results

This section presents the quantitative findings from our new coupled model. We begin by verifying that the candidate equilibrium computed with the standalone model is consistent with the behavior of the fully coupled system. We then analyze the secular trends in temperature and GDP projected by the model, both globally and regionally. Finally, we investigate short-term fluctuations in global and regional GDP, highlighting the deviations from long-term trends that arise from internal variability in NorESM2–DIAM.

**5.1    Assessing the candidate equilibrium**

We initialize cumulative emissions in 1990 to match the historical NorESM2 input (Eyring et al., 2015). Figure 8 presents the equilibrium path of cumulative emissions in the standalone model (absent internal variability), alongside historical emissions





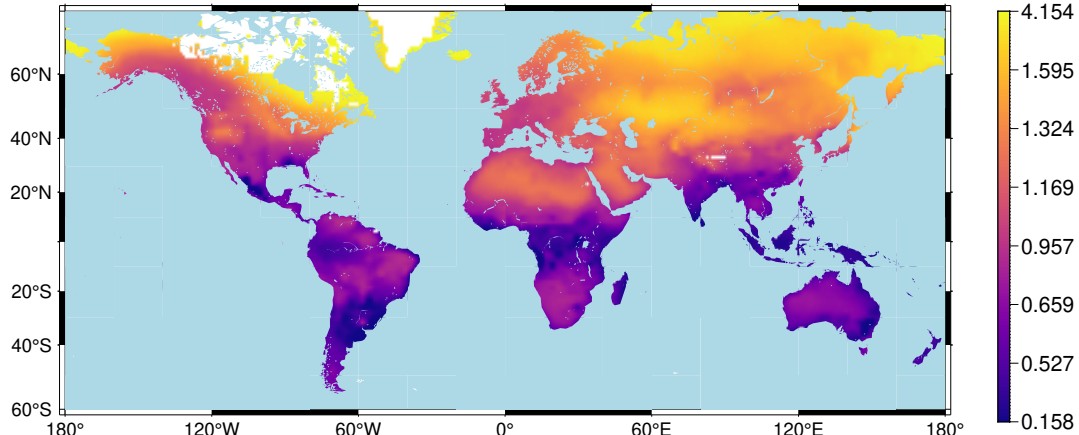

**Figure 7.** Regional warming in response to a $1^\circ$C increase in area-weighted average temperature (over regions with economic activity) based on the statistical temperature forecast approach. The color bar is not linear; instead each increment in the color bar represents the same number of regions.

through 2014 and four Shared Socioeconomic Pathway (SSP) projections thereafter. From 1990 to 2024, the standalone model closely tracks observed cumulative emissions. Beyond 2024, its trajectory aligns most closely with SSP3-7.0, before flattening and ultimately ending the century between SSP2-4.5 and SSP3-7.0.

Figure 9 shows the difference in annual and cumulative emissions between the standalone model and those generated by NorESM2–DIAM. In the coupled model, internal variability in regional temperatures causes realized regional emissions—which depend on actual rather than expected temperatures—to diverge from expectations. These deviations do not cancel out across regions: even at the global level, emissions often differ from the standalone model by several percentage points in absolute value, as shown by the black line in Fig. 9. However, because annual flows are small relative to the stock of cumulative emissions, the coupled model's cumulative emission path remains close to that of the standalone model, differing by no more than 0.4% (gray line in Fig. 9). As a result, agents' forecasts of future expected temperature—based on the standalone model's cumulative emissions path—are highly accurate: accounting for the minor deviations in the coupled model would change forecasts only marginally. Taken together with the assumption that the AR(1) process describing deviations of regional temperature from its expected path is invariant to global warming (see Section 2.3.1), this result supports the conclusion that the behavior of the coupled model provides a reasonable approximation to an exact economic equilibrium.

## 5.2 Global change

Figure 10 (a) displays the path for population-weighted global temperature in NorESM2-DIAM, together with the trend path from the standalone model. (We use population-weighted temperature to focus on regions where people live; recall that regional population shifts over time so that the population weights change over time.) The global temperature in NorESM–DIAM tracks the trend path, providing additional evidence that the behavior of NorESM–DIAM aligns with the candidate equilibrium from





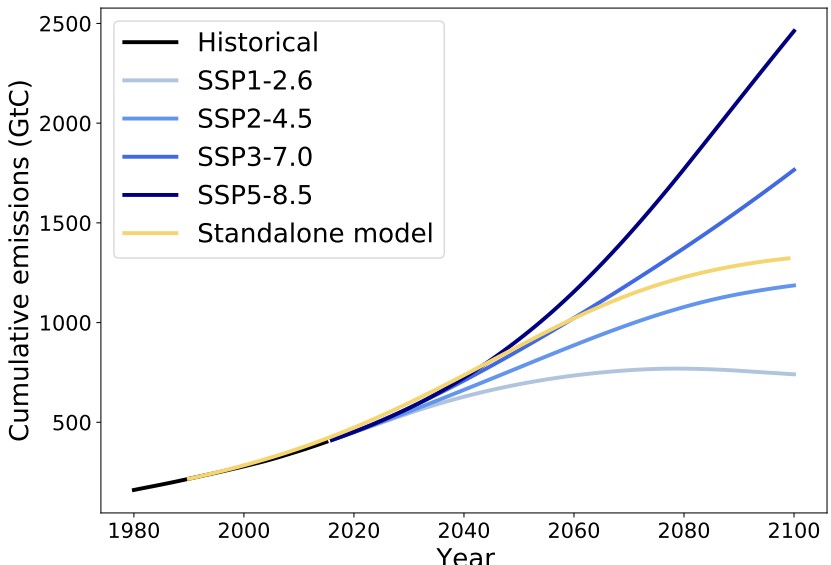

**Figure 8.** Cumulative $CO_2$ emissions (excluding land–use change) since 1850 for the years 1980 to 2100 measured in GtC. The black line is the emissions used by NorESM2 for historical simulations, based on estimated historical emissions, and goes until 2014. The four blue lines are the emissions used in the four most common emission scenarios, starting in 2014. The yellow line is cumulative emissions from the standalone model, which starts from the historical cumulative emissions in 1990.

the standalone model. There are substantial variations in global temperature around this trend path, driven by internal variability in NorESM2. Comparing with Fig. 9, there is a positive relationship between deviations of global emissions and deviations of global temperature from their respective trend paths: higher temperatures tend to reduce global productivity (as we discuss
further below), in turn leading to lower energy use and fewer emissions.

Note that the NorESM2–DIAM simulation begins from a relatively cold temperature compared with the trend path in the standalone model. This discrepancy reflects two factors: the historical simulation used to initialize NorESM2–DIAM is relatively cool in 1990, and the statistical temperature forecast provides a less precise fit at low levels of cumulative emissions (see Fig. 6). Nevertheless, the statistical temperature forecast captures NorESM2's behavior well overall.
The corresponding change in global GDP is shown in Fig. 10 (b), expressed as a percentage relative to 1990. To isolate the effects of climate change and population dynamics, we remove the underlying constant growth rate of 1.5% driven by the exogenous component of productivity. As expected, GDP in NorESM2-DIAM closely tracks the standalone model, consistent with the alignment in temperatures between the two models. Global GDP rises until around 2040, reaching about 35% above 1990 levels, before beginning to decline. The initial increase reflects population growth, which peaks in the 2080s, while the
subsequent downturn results from both population shifts and the impacts of climate change, as we discuss further below.



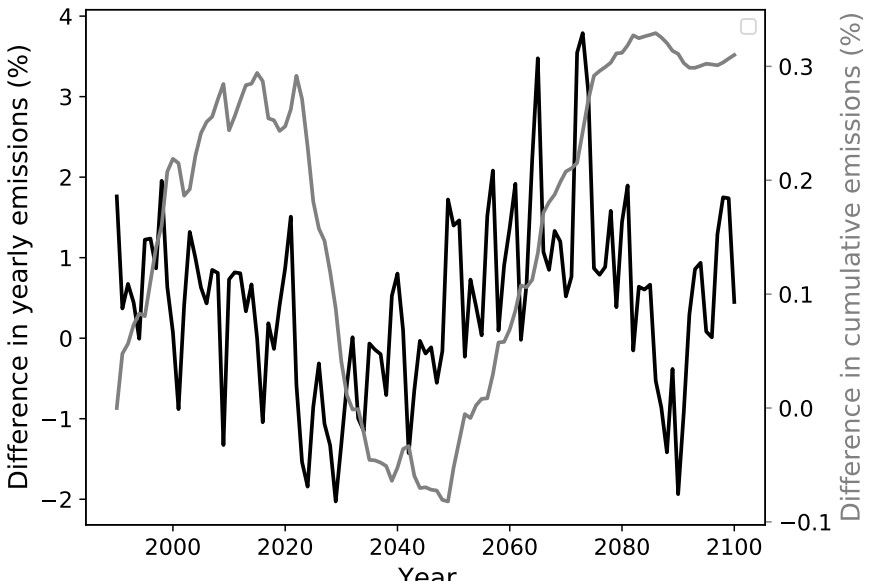

**Figure 9.** Percentage difference in annual emissions (black) and cumulative emissions grey) between the coupled NorESM2-DIAM simulation and the DIAM standalone.

Finally, Fig. 10 (c) displays the change in global GDP per capita since 1990 (again with the underlying growth trend of 1.5% removed). Once again, NorESM2–DIAM tracks the trend in the standalone model closely, and in both models global GDP per capita declines by about 35% by 2100.

To understand this decline, Fig. 10 (c) also displays the results of two counterfactual experiments using the standalone model, one in which the climate changes over time but regional population does not, and another in which regional population changes over time but the climate does not. These experiments reveal that most of the decline in GDP per capita (about 64%) can be attributed to shifts in regional population. In particular, as shown in Fig. 2, according to our projections, the distribution of global population changes strikingly over time, with population tending to grow (shrink) in regions that are relatively poor (rich) in 1990, as measured by the initial value of the exogenous component of productivity (which in turn reflects differences in regional GDP per capita in 1990). Even as global GDP increases initially (due to a growing global population), global GDP per capita declines throughout because poor regions are becoming relatively more populated.

Climate change alone, in turn, causes a decline of about 8% in GDP per capita by 2100, or about 23% of the total decline. This decline is a quantitative measure of the global damages from climate change generated by NorESM–DIAM when the global (population-weighted) temperature increases by close to 3.5°C. These global damages are larger than in the most recent version of Nordhaus's DICE model (see Barrage and Nordhaus (2024) in which damages are about 4% of global GDP at 3.5°C of warming) but in line with other estimates in the literature (see, for example, Rennert et al., 2022). In NorESM2–DIAM,





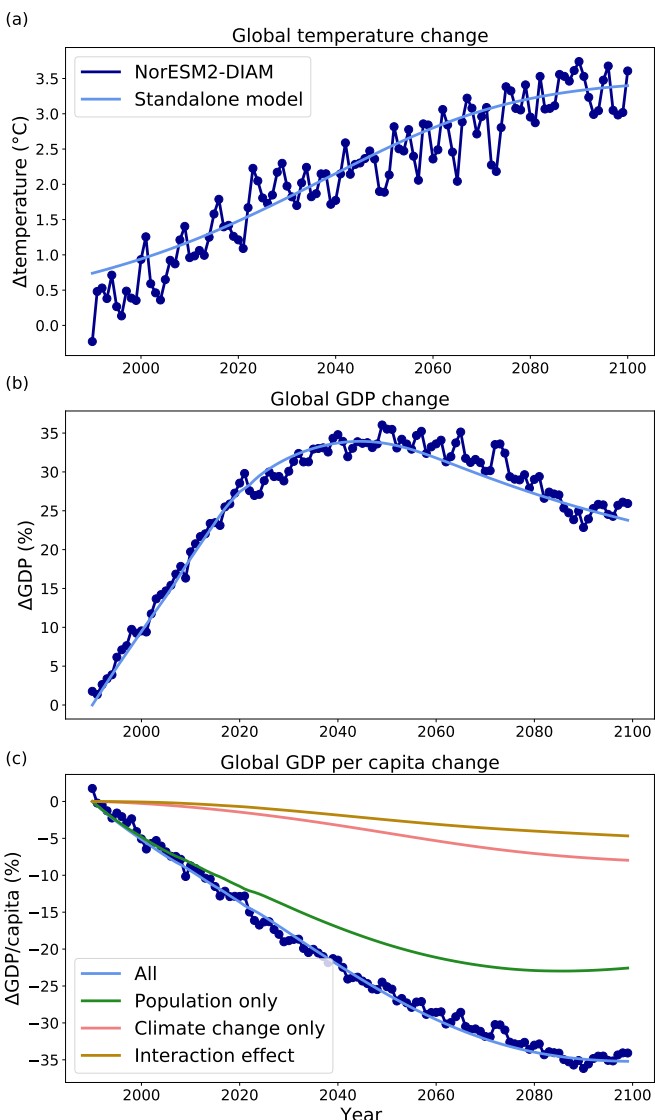

**Figure 10.** Global mean values of (a) population weighted temperature change from pre-industrial, (b) percentage change in GDP since 1990, and (c) percentage change in GDP per capita since 1990. For GDP and GDP per capita the exogenous growth is removed. The solid line is the value from the DIAM standalone model, while the dashed line is the calculated value from the coupled NorESM2-DIAM. Panel (c) also shows the decomposition of GDP per capita into contributions from population only (green), climate change only (red), and interaction effects (yellow).





global damages depend critically on the regional damage function $D$ (see Fig. 3), and the specific calibration of this function that we use here can then be viewed as a reasonable one in the sense that it generates quantitatively reasonable global damages (see Krusell and Smith (2022) and Bjordal et al. (2022) for a thorough discussion).

The rest of the decline in global GDP per capita (about 13%) can be attributed to an interaction effect between climate change and population shifts, as shown in Fig. 10 (c): in particular, population tends to shift over time not only to poorer regions but also to hotter regions, that is, regions whose initial temperatures in 1990 are to the right of the optimum temperature in see Fig. 3 and which therefore experience greater damages from climate change than cooler regions. (The latter may even experience gains from climate change if they are cool enough, a point to which we return in Section 5.3 below.)

As discussed in section 4.1, in our calibration the exogenous component of productivity has the same constant growth rate in all regions. A more realistic calibration might allow poorer regions to grow faster initially as they catch up to the global technology frontier. Such a calibration would lead to a different decomposition than the one displayed in Fig. 10 (c), likely reducing the role of population shifts in causing declines in global GDP per capita. It is entirely feasible in our methodology to allow the growth rate of the exogenous component of productivity to vary across time and space, but we leave this to future
work.

## 5.3   Regional change

We now turn to patterns of regional change. Figure 11 shows, for both the standalone model and the fully-coupled model, projected changes in regional temperature from 1990 to 2040 and 2090 in response to global warming. Consistent with Fig. 7, regions in high northern latitudes warm the most, reflecting Arctic amplification (Meredith et al., 2019). Differences between
the standalone and NorESM2–DIAM simulations reflect internal variability in NorESM2–DIAM, which is suppressed in the standalone model.

The maps in Fig. 12 show how the percentage change in GDP per capita is distributed corresponding to the temperature maps in Fig. 11. Note that these are relative to the underlying trend path growing at 1.5% year (driven by growth in the exogenous component of productivity). There are large differences across regions, with many regions' economies growing
relative to trend growth, often by large amounts, and other regions' economies shrinking. Moreover, the differences across regions increase substantially over time, with the percentage changes in regional GDP from 1990 to 2090 displaying much more spatial variation than the percentage changes from 1990 to 2040.

These patterns, in turn, reflect how regional productivity varies over time as the global climate warms. Cool regions, located to the left of the optimum temperature in the damage function $D$ in 1990 (see Fig. 3), warm over time, leading to increases
in productivity as their temperatures move towards the optimum temperature. Initially warm regions, by contrast, decline in productivity as their temperatures move away from the optimum temperature.

As for regional temperatures, the percentage changes in regional GDP differ across the standalone model and NorESM2–DIAM, reflecting internal variability in NorESM2–DIAM: realized regional temperatures fluctuate relative to expected temperature, leading to fluctuations in productivity and hence GDP. We discuss this variability in Section 5.4 below.




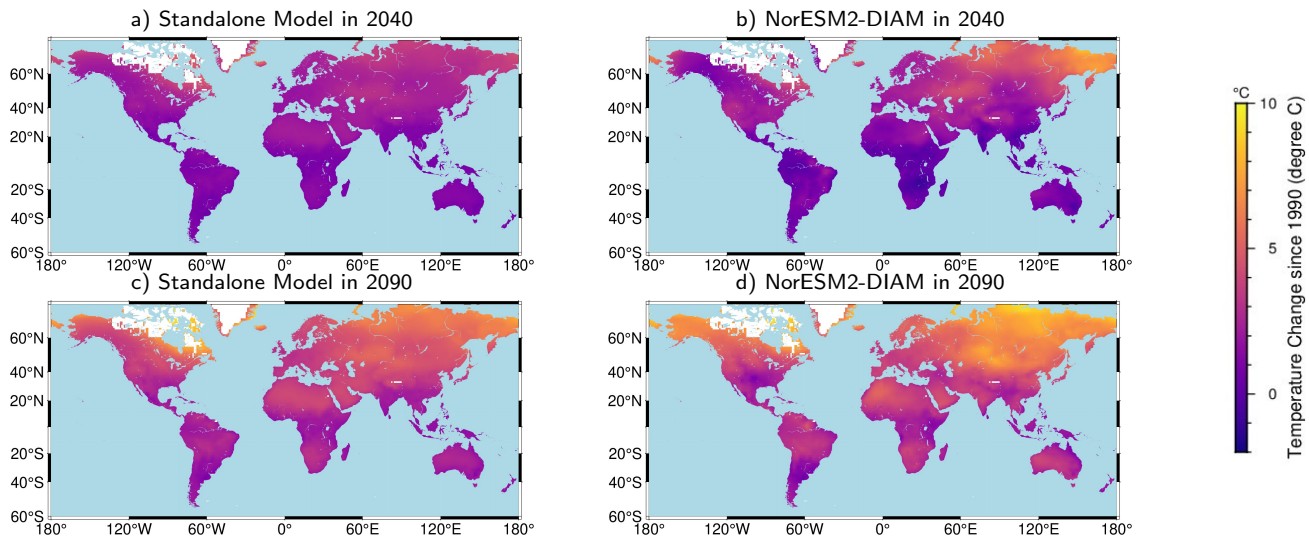

**Figure 11.** Maps of temperature change from pre-industrial for years 2040 and 2090, for the DIAM standalone and the coupled NorESM2-DIAM. a) is the DIAM standalone for year 2040, b) is NorESM2-DIAM for year 2040, c) is the DIAM standalone for 2090, and d) is NorESM2-DIAM for year 2090.

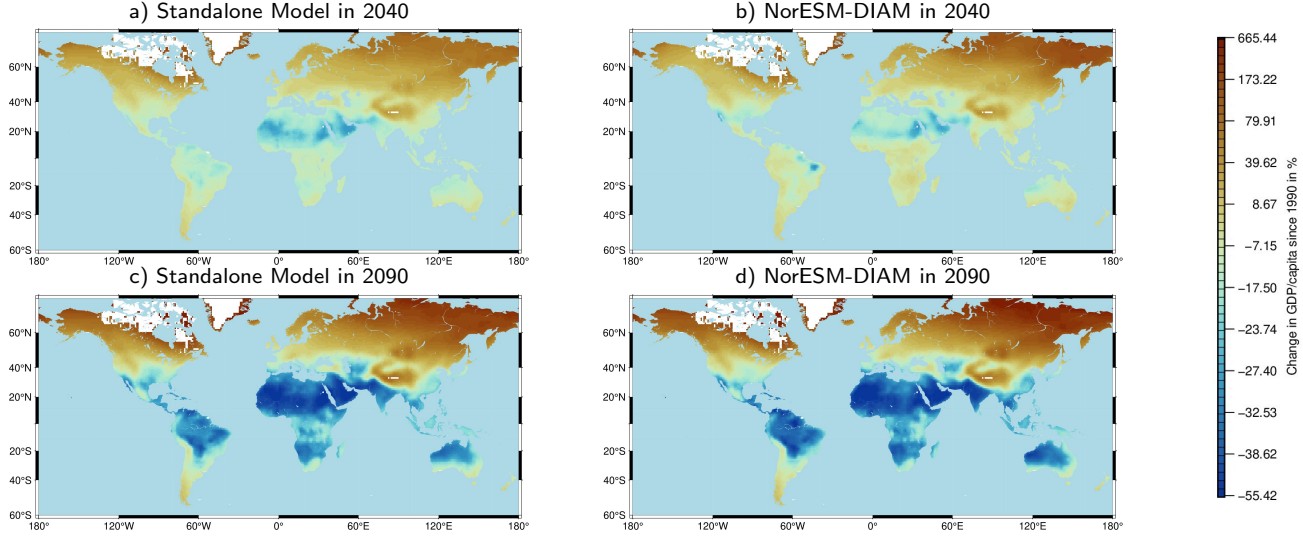

**Figure 12.** Maps of percentage change in regional GDP per capita (relative to the underlying trend path growing at 1.5% per year) from 1990 to 2040 and 2090. a) is the standalone model for year 2040, b) is NorESM2-DIAM for year 2040, c) is the standalone model for 2090, and d) is NorESM2-DIAM for year 2090. Note that the color bar is not linear; instead each increment in the color bar represents the same number of regions.





In the maps in Fig. 12, GDP per capita increases in large areas, seeming to suggest that global GDP per capita increases over time, rather than decrease as shown in Fig. 10 (c). But it is important to note that most of the regions in which GDP per capita increases have small populations and consequently contribute little to global GDP. To see this more clearly, Fig. 13 (a) and (b) aggregates regions into countries and shows the percentage change in each country's temperature and GDP per capita from the last decade of the 20th century (1990-1999) to the last decade of the 21st century (2090-2099). These figures reveal

that for most, though not all, countries, GDP per capita declines over the century. Moreover, global GDP per capita (see the circle labeled "GLOBAL") has a larger decrease than most individual countries.

To better understand this finding, Fig. 13 (a) emphasizes the effect of population. Here, the size of each circle corresponds to initial population and the color of each circle corresponds to changes in population. Many of the countries with a large decrease in GDP per capita also have a large increase in population, whereas for many of the countries in which GDP per

capita increases, population decreases. Consequently, global GDP per capita falls by more than it does in many individual countries.

Figure 13 (b) is the analogue of Fig. 13 (a), but here the size of each circle corresponds to initial temperature (i.e., the decadal average for 1990-1999) and the color of each circle corresponds to GDP per capita in 1990-1999. As is the case for individual regions, in warm countries GDP per capita decreases over time (relative to the underlying trend), while in colder

countries it increases. This figure shows clearly that most poor countries (i.e., those represented by smaller circles) experience large damages from climate change in the sense that GDP per capita falls substantially in them.

## 5.4    Variability in productivity and GDP

Internal variability in NorESM2–DIAM leads to large fluctuations in regional temperatures around their trend paths, as quantified in the estimated coefficients of the AR(1) process that agents use to make forecasts of future regional temperatures (see

Section 2.3.1). Likewise, the global temperature fluctuates substantially around its trend path, as illustrated in Fig. 10 (a).

These temperature fluctuations, in turn, lead to quantitatively significant fluctuations in productivity and hence in regional and global GDP. Figure 14 shows that in NorESM2–DIAM, the standard deviation of (annual) regional GDP, expressed as a percentage relative to its trend, ranges from near zero to 33%, with most values lying between 2% and 10%. The large spatial heterogeneity in this measure of GDP volatility has two sources. First, there is substantial spatial heterogeneity in the volatility

of regional temperature itself. Second, average regional temperatures vary greatly across space, so that regions are located at very different points along the inverse $U$-shaped damage function (see Fig. 4.5) determining regional productivity. For a given amount of volatility in regional temperature, regions near the peak of the damage function experience smaller fluctuations in productivity than regions either to the left or right of the peak where the slope of the damage function is larger (in absolute value).

Figure 13 (c) illustrates this variability at the country level. Specifically, this figure shows the differences between NorESM2-DIAM and the standalone model, i.e., the difference between the change from 1990-1999 to 2090-2099 in the two models. Averaging over a decade dampens a considerable amount of the internal variability in NorESM–DIAM. Nonetheless, even over this longer horizon, internal variability still leads to quantitatively important variations in GDP per capita. (Note that the





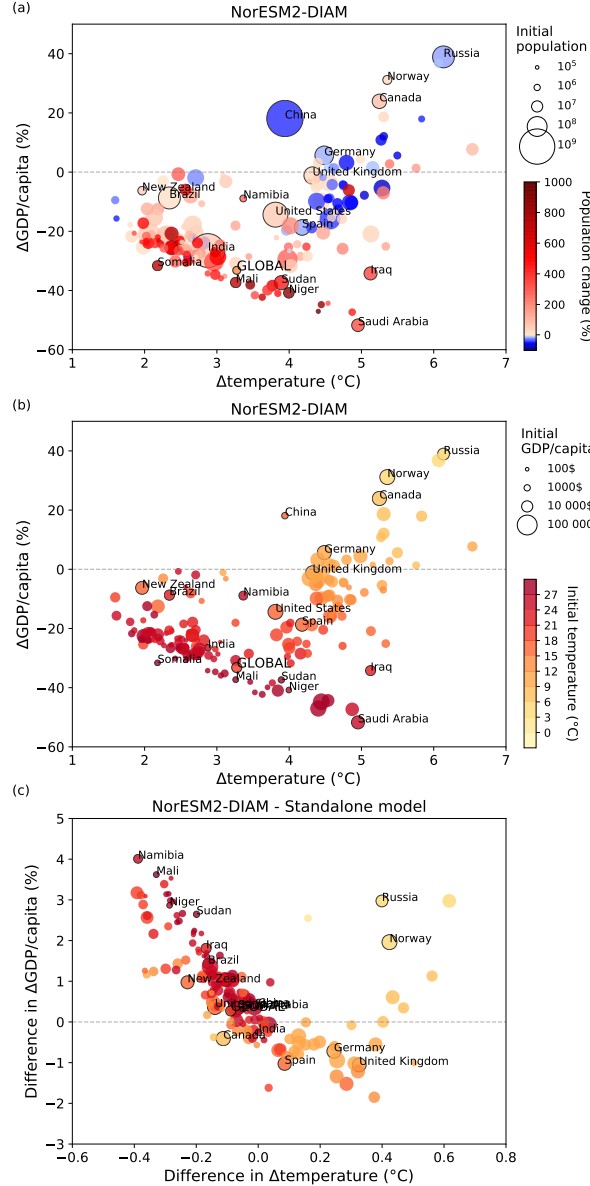

**Figure 13.** Country-level change in temperature and GDP per capita in 2090-2099 compared to 1990-1999. a) and b) show each country's percentage change in decadal GDP per capita on the y-axis against decadal population-weighted temperature change on the x-axis as calculated by NorESM2-DIAM. c) show the differences between NorESM2-DIAM and the standalone model for the change in temperature and GDP per capita. In a) each country's circle, as well as the global mean, is colored based on the percentage change in population (2090-2099 compared to 1990-1999), and the size indicates the population in year 1990. In b) and c) the top row, each country's symbol, as well as the global mean, is colored based on the 1990-1999 population-weighted temperature, and the size indicates the GDP per capita average over 1990-1999.





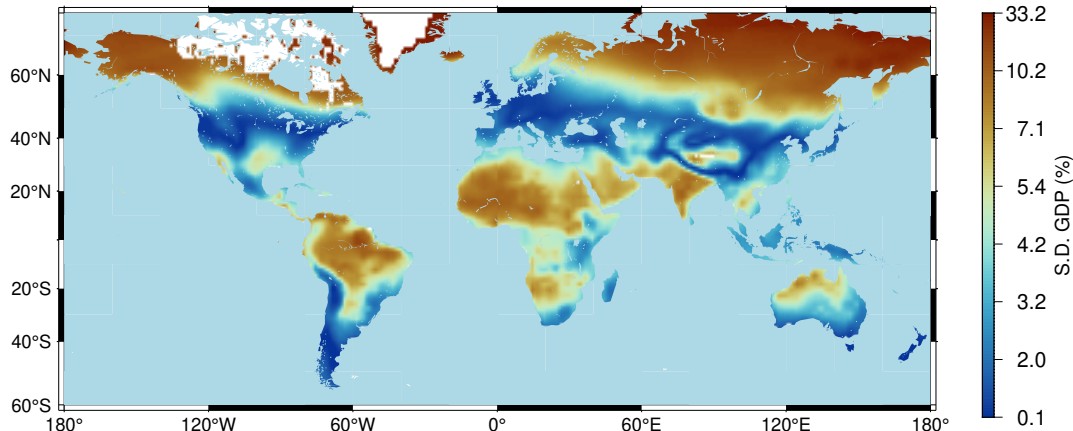

**Figure 14.** Standard deviation of regional GDP (expressed as a percentage relative to trend).

contribution from population changes is not relevant here, as the two have the same population changes.) For example, in a cold

country that experiences a higher-than-normal temperature, productivity increases, leading to an increase in GDP per capita in the short run. Similarly, in a hot country that experiences a lower-than-normal temperature, productivity also increases, leading again to an increase in GDP per capita. By contrast, in a cold country that experiences a lower-than-normal temperature (or in a hot country that experiences a higher-than-normal temperature), productivity and GDP per capita fall in the short run.

Finally, as shown in Fig. 10 (b), global GDP itself experiences large fluctuations relative to its trend, about 1% in percentage

terms. Evidently, even though the number of regions is large, deviations of regional GDP do not wash out in the aggregate, for two reasons. First, regional temperatures are correlated in space. Second, economic activity is highly concentrated in space. As a result, the effective number of regions is much smaller than 19,000, slowing down the action of the law of large numbers.

## 6   Discussion

We have developed a coupled model consisting of two main components—an IAM and an ESM—that exchange information

every year at a gridded level. We find, like Krusell and Smith (2022), Bjordal et al. (2022), and Cruz and Rossi-Hansberg (2024), that changes to GDP per capita induced by global warming vary greatly across regions, underscoring again the importance of making regional assessments of economic impacts. In line with previous research (Kotz et al., 2021; Kikstra et al., 2021; Waidelich et al., 2024; Kotz et al., 2024), we also find that internal variability—which is ignored in most IAMs—can be important for assessing the economic impacts of global warming, both annually and on longer time scales.

The quantitative results depend critically on the shape of the regional damage function. We show that our calibration of the damage function generates aggregate damages in line with existing estimates, but it is important in future work to provide stronger empirical foundations for this function. Specifically, we plan to evaluate the quantitative effects of different damage functions, including those that depend on additional climate and weather variables beyond annual mean temperature. The





damage function we use here also assumes implicitly that a permanent change in climate (such as a permanent increase in average annual temperature) has the same economic impacts as a transitory change in weather (such as relatively hot or cold year), ignoring possible adaptation to changes in climate. We hope to address this shortcoming in future work too.

In the prototype model implemented here, NorESM2 and DIAM exchange only regional temperatures and emissions. But the NorESM2-DIAM framework opens many possibilities for future model development. For example, we can easily extend our methodology to include additional climate variables which have been shown to be important for assessing the economic damages of climate change. (Waidelich et al., 2024; Kotz et al., 2024). NorESM2 already provides information on a plethora of climate and weather variables. Given a damage function that depends on (a subset of) these variables, extending the methodology simply requires incorporating these additional variables into the statistical model that agents use to forecast future regional damages.

Another important question is whether climate change affects the growth rate of economic activity, rather than merely shifting the level of activity as in our prototype model. This issue remains unsettled (see, e.g., Dell et al., 2012; Howard and Sterner, 2017; Burke et al., 2015), but again our framework can be readily extended to accommodate such effects by modifying the damage function accordingly.

The economic model (DIAM) in the prototype model developed here is relatively simple, with several important limitations that could be important for assessing the spatial effects of climate change. These include constant exogenous productivity growth across time and space, no capital mobility, and exogenous population changes. We also limit attention to $CO_2$, though including forcings from other greenhouse gases would permit better a better representation of how the climate changes in response to economic decisions. However, many of these limitations (with the possible exception of migration in response to climate change) can be easily incorporated into the existing framework. The advantage of starting with a simple model is that its output is relatively easy to understand and interpret. This model can also serve as a baseline against which future model versions with more complex connections between climate and the economy can be compared.

An important limitation of NorESM2-DIAM, compared to existing IAMs, is its computational cost. While IAMs such as DICE or PAGE can be run in a matter of minutes or less (e.g. Moore et al., 2018) on any computer, full ESMs—and consequently NorESM2-DIAM—take hours or days on a supercomputer. Therefore, we must limit the number and length of model simulations. However, the computational cost is close to that of running an ESM, so running NorESM2 with an economic module has a negligible effect on the overall run time. In principle, NorESM2-DIAM can be used for all the same experiments as IAMs. However, due to the computational cost, it makes most sense to use NorESM2-DIAM to answer questions where a good representation of the climate system and the carbon cycle is important, such as how extreme weather events and internal variability affect economic outcomes.

In addition to the NorESM2-DIAM model, we now have a simple representation of NorESM2's climate in the standalone version of DIAM. The standalone model is computationally inexpensive relative to the full coupled model, so it can be useful when speed is important, for example, to perform many different simulations or very long simulations. The standalone model can also be used as a guide to what simulations it would be worthwhile to run in the full model. Finally, in cases where we have already performed simulations of the full model with similar emission trajectories, we could use NorESM2 data




for regional temperatures (and possibly other variables) as an input to the standalone model, in a sort of offline coupling:
different combinations of economic growth, greening, and policy can in some cases deliver similar paths for emissions and
consequently temperatures, yet still have very different economic impacts. Finally, the simple model could also be useful for
teaching purposes.

## 7    Conclusions

In conclusion, the NorESM2-DIAM framework successfully couples a state-of-the-art Earth System Model and a cost-benefit
Integrated Assessment Model with high geographical resolution. The new model exchanges temperature and $CO_2$ emissions
on a yearly basis on a regional gridded level, generating dynamic emissions trajectories. The results highlight the importance
of spatial and temporal variability for economic outcomes as well as the wide range of outcomes among regions.

A caveat to keep in mind is that these are early results using a coupled model that is, to our knowledge, the first of its
kind. The quantitative results presented here should therefore be interpreted with caution. There are still large uncertainties,
particularly regarding the proper form of the damage function, and there are several important features (see the discussion in
Section 6) that we would like to add to the model in order to assess their quantitative importance. Nevertheless, our results
demonstrate how the two components of the coupled model work together and the framework we develop here is a good starting
point for future model development, opening up a wide range of new opportunities for more comprehensive and sophisticated
simulations of climate-economy interactions.

*Code and data availability.* NorESM2 is openly available to download from https://github.com/NorESMhub/NorESM. Here we have used
the following release: release-noresm2.0.9. The input data used by the model are available from https://noresm.org/inputdata/. See also
developers group (2021) for more information. The code for standalone DIAM, coupling, and calculations are available from https://
github.com/jennybj/coupling_noresm2_diam, using the release: release-v1.0.0 (https://doi.org/10.5281/zenodo.17176880). Raw output from
NorESM2 is available from https://doi.org/10.11582/2025.31ney5y8, output from the coupling is available from https://doi.org/10.11582/
2025.90v981qk, and output from standalone DIAM is available from
https://www.dropbox.com/scl/fo/mm9utacdrk42fmzv6juh4/AIUr4sSBMterks3Tjsd3YEU?rlkey=plm6rqom86dqasan7cf13ge0r&st=hcc7c3e3&
dl=0. An overview of the availability of both input and output data is also specified in the Github repository's README. The README also
contains instruction to replicate the results, and running the model.

## Appendix A

## A1    Introduction

Sections A2 to A6 of this Appendix explain how we solve the regional dynamic programming problems. Section A7 gives
details on how we execute a forward simulation using NorESM2.





## A2 Setup

Time is discrete and starts in year 0, corresponding to the real-world year $R$. Agents in region $i$ assume that their temperature
in year $t$, $T_{it}$, is given by:

$$T_{it} = \overline{T}_{it} + z_{it}, \tag{A1}$$

where $\overline{T}_{it}$ is region $i$'s expected temperature in year $t$ and $z_{it}$ is a region-specific random shock to regional temperature.
Assume that $\overline{T}_{it} = \overline{T}_i + \gamma_{i1}S_t + \gamma_{i2}S_t^2$, where $\overline{T}_i$ is pre-industrial temperature in region $i$ and $S_t$ is cumulative global carbon
emissions (since the pre-industrial era) at the beginning of year $t$. Assume that $z_{it}$ follows an AR(1) process:

$$z_{it} = \rho_i z_{i,t-1} + \epsilon_{it}, \tag{A2}$$

where $\{\epsilon_{it}\}_{t=0}^\infty$ is an i.i.d. (independent and identically distributed) sequence of random variables with a $N(0,\sigma_i^2)$ distribution.

Let $k_{it}$ be the physical capital stock in region $i$ at the beginning of year $t$, let $\omega_{it}$ denote wealth in region $i$ at the beginning of
year $t$, and let $x_{it}$ denote energy use (measured in BTUs) in region $i$ during year $t$. Let $\delta$ be the rate at which capital depreciates.
Let population in region $i$ in year $t$, $N_{it}$, evolve according to $N_{i,t+1} = (1 + g_{i,t+1}^N)N_{it}$, where $N_{i0}$ is a given number and
$\{g_{it}^N\}_{t=1}^\infty$ is an exogenous sequence.

Let $\psi\phi_{it}$ be carbon emissions (in GtCs) per BTU in region $i$ in year $t$ and let $e_{it}$ be carbon emissions in region $i$ in year $t$,
i.e., $e_{it} = \psi\phi_{it}x_{it}$. Then global emissions in year $t$ are equal to:

$$E_t \equiv \sum_{i=1}^M e_{it},$$

where $M$ is the number of regions. Then $S_{t+1} = S_t + E_t$ for $t \geq 0$, where $S_0$ is cumulative emissions (since the pre-industrial
era) through the beginning of real-world year $R$. Alternatively, for $t \geq 1$,

$$S_t = S_0 + \sum_{s=0}^{t-1} E_s.$$

Each region takes as given the sequence $\{S_t\}_{t=0}^\infty$. Using (A1), this sequence in turn determines the sequence $\{\overline{T}_{it}\}_{t=0}^\infty$.

Finally, assume that $S_t = S^*$ for $t \geq t_1$ (i.e., starting in year $t_1$, $\phi_{it} = 0$ for all $i$, so that there are no further carbon emissions).

## A3 Dynamic program of a typical region

Each region $i$ solves the following dynamic programming problem, where $\omega_{it}$ is aggregate wealth in region $i$ at the beginning
of year $t$ and $k_{it}$ is the aggregate capital stock in region $i$ at the beginning of year $t$:

$$v_t^i(\omega_{it}, z_{it}, N_{it}, A_{it}) =$$

$$\max_{k_{i,t+1}} \left[ N_{it} U\left(\frac{\omega_{it} - k_{i,t+1}}{N_{it}}\right) + \beta\mathbb{E}_t\left(v_{t+1}^i(\omega_{i,t+1}, z_{i,t+1}, N_{i,t+1}, A_{i,t+1})\right) \right],$$





subject to eq. (A1), (A2), the borrowing constraint $k_{i,t+1} \geq 0$, and the law of motion for wealth:

$$\omega_{i,t+1} = \max_{x_{i,t+1}} \left( F\left(k_{i,t+1}^{\alpha}\ell_{i,t+1}^{1-\alpha}, x_{i,t+1}\right) - p_i x_{i,t+1} \right) + (1-\delta)k_{i,t+1}, \tag{A3}$$

where $\ell_{it} = N_{it}A_{it}D(T_{it})$ is aggregate efficiency units of labor in region $i$ in year $t$, $p_i$ is the price of a unit of energy (expressed in units of the final consumption good), $D(T_{it})$ is a nonnegative, inverse $U$-shaped function with a unique maximum at $D(T^*) = 1$, and the sequence $\{A_{it}\}_{t=0}^{\infty}$ obeys: $A_{i,t+1} = (1 + g_{i,t+1}^A)A_{it}$, where $A_{i0}$ is a given number and $\{g_{it}^A\}_{t=1}^{\infty}$ is an exogenous sequence. Note that the value function in region $i$, $v_t^i$, implicitly depends on the region-specific parameters $\overline{T}_i$, $\rho_i$,

$\gamma_{i1}$, $\gamma_{i2}$, and $\sigma_i^2$; the region-specific sequences of growth rates $\{g_{it}^N\}_{t=1}^{\infty}$ and $\{g_{it}^A\}_{t=1}^{\infty}$; and the common sequence $\{S_t\}_{t=0}^{\infty}$.

Assume finally that, for $t \geq t_2 > t_1$, $g_{it}^N = g^N$ and $g_{it}^A = g^A$, i.e., the growth rates of population and exogenous technological progress are constant across time and space starting in year $t_2$.

## A4 Detrending the dynamic program

For any variable $y_{it}$, define the scaled variable

$$\hat{y}_{it} \equiv \frac{y_{it}}{N_{it}A_{it}D(\overline{T}_{it})}.$$

Because the production function, $F$, is assumed to have constant returns to scale in its two arguments, the law of motion for

wealth (eq. A3) can be rewritten:

$$\hat{\omega}_{i,t+1} = \max_{\hat{x}_{i,t+1}} \left( F\left(\hat{k}_{i,t+1}^{\alpha}\left(d_{t+1}^i(T_{i,t+1})\right)^{1-\alpha}, \hat{x}_{i,t+1}\right) - p_i\hat{x}_{i,t+1} \right) + (1-\delta)\hat{k}_{i,t+1}, \tag{A4}$$

where

$$d_t^i(\cdot) \equiv \frac{D(\cdot)}{D(\overline{T}_{it})}.$$

Assume that $U(c) = c^{1-\gamma}$. Then

$$U\left(\frac{\omega_{it} - k_{i,t+1}}{N_{it}}\right) = \left(A_{it}D(\overline{T}_{it})\hat{\omega}_{it} - \frac{N_{i,t+1}A_{i,t+1}}{N_{it}A_{it}}A_{it}D(\overline{T}_{i,t+1})\hat{k}_{i,t+1}\right)^{1-\gamma}$$

$$= \left(A_{it}D(\overline{T}_{it})\right)^{1-\gamma}\left(\hat{\omega}_{it} - (1+g_{i,t+1}^N)(1+g_{i,t+1}^A)d_t^i(\overline{T}_{i,t+1})\hat{k}_{i,t+1}\right)^{1-\gamma}.$$

Guessing that $v_t^i(\omega_{it}, z_{it}, N_{it}, A_{it}) = N_{it}\left(A_{it}D(\overline{T}_{it})\right)^{1-\gamma}\hat{v}_t^i(\hat{\omega}_{it}, z_{it})$, rewrite region $i$'s dynamic program as:

$$N_{it}\left(A_{it}D(\overline{T}_{it})\right)^{1-\gamma}\hat{v}_t^i(\hat{\omega}_{it}, z_{it}) =$$

$$\max_{\hat{k}_{i,t+1}, \hat{x}_{i,t+1}} \left[ N_{it}\left(A_{it}D(\overline{T}_{it})\right)^{1-\gamma}\left(\hat{\omega}_{it} - (1+g_{i,t+1}^N)(1+g_{i,t+1}^A)d_t^i(\overline{T}_{i,t+1})\hat{k}_{i,t+1}\right)^{1-\gamma} \right.$$

$$\left. \beta\mathbb{E}_t\left(N_{i,t+1}\left(A_{i,t+1}D(\overline{T}_{i,t+1})\right)^{1-\gamma}\hat{v}_{t+1}^i(\omega_{i,t+1}, z_{i,t+1})\right) \right],$$

subject to eq. (A1), (A2), (A4), and the borrowing constraint. Simplify this equation to get:

$$\hat{v}_t^i(\hat{\omega}_{it}, z_{it}) = \tag{A5}$$





$$\max_{\hat{k}_{i,t+1},\hat{x}_{i,t+1}} \left[ \left( \hat{\omega}_{it} - (1+g^N_{i,t+1})(1+g^A_{i,t+1})d^i_t(\overline{T}_{i,t+1})\hat{k}_{i,t+1} \right)^{1-\gamma} + \right.$$

$$\left. \beta(1+g^N_{i,t+1})(1+g^A_{i,t+1})^{1-\gamma} \left( d^i_t(\overline{T}_{i,t+1}) \right)^{1-\gamma} \mathbb{E}_t \left( \hat{v}^i_{t+1}(\hat{\omega}_{i,t+1},z_{i,t+1}) \right) \right],$$

subject to eq. (A1), (A2), (A4), and the borrowing constraint.

### A5   Solving the dynamic program backwards

Starting in year $t_2$, region $i$'s dynamic programming problem then has a time-invariant solution and simplifies to:

$$\hat{v}^i(\hat{\omega}_{it},z_{it}) = \max_{\hat{k}_{i,t+1},\hat{x}_{i,t+1}} \left[ \left( \hat{\omega}_{it} - (1+g^N)(1+g^A)\hat{k}_{i,t+1} \right)^{1-\gamma} + \right. \tag{A6}$$

$$\left. \beta(1+g^N)(1+g^A)^{1-\gamma} \mathbb{E}_t \left( \hat{v}^i(\hat{\omega}_{i,t+1},z_{i,t+1}) \right) \right],$$

subject to eq. (A1), (A2), (A4), and the borrowing constraint, where now

$$\overline{T}_{it} = \overline{T}^*_i \equiv \overline{T}_i + \gamma_{i1}S^* + \gamma_{i2}\left(S^*\right)^2.$$

The first step in solving problem (A5) is to find the function $\hat{v}^i(\hat{\omega}_{it},z_{it})$ that solves equation (A6).

Next, for $t = t_1,\ldots,t_2-1$, problem (A5) simplifies to:

$$\hat{v}^i_t(\hat{\omega}_{it},z_{it}) = \tag{A7}$$

$$\max_{\hat{k}_{i,t+1},\hat{x}_{i,t+1}} \left[ \left( \hat{\omega}_{it} - (1+g^N_{i,t+1})(1+g^A_{i,t+1})\hat{k}_{i,t+1} \right)^{1-\gamma} + \right.$$

$$\left. \beta(1+g^N_{i,t+1})(1+g^A_{i,t+1})^{1-\gamma} \mathbb{E}_t \left( \hat{v}^i_{t+1}(\hat{\omega}_{i,t+1},z_{i,t+1}) \right) \right],$$

subject to eq. (A1), (A2), (A4), and $\overline{T}_{it} = \overline{T}^*_i$ for all $t$. Note that

$$\hat{v}^i_{t_2}(\hat{\omega}_{i,t_2},z_{i,t_2}) = \hat{v}^i(\hat{\omega}_{i,t_2},z_{i,t_2}),$$

where the latter function was computed in the first step. The second step in solving problem (A5) is to find, working backwards
from $t = t_2 - 1$ to $t = t_1$, the sequence of functions $\hat{v}^i_t(\hat{\omega}_{it},z_{it})$ that solves eq. (A7) for $t = t_1,\ldots,t_2-1$.

The third and final step is then to iterate backwards on eq. (A5), from $t = t_1 - 1$ to $t = 0$, to find the sequence of functions $\hat{v}_{it}(\hat{\omega}_{it},z_{it})$ that solves eq. (A5) for $t = 0,\ldots,t_1-1$.

A byproduct of solving problem (A5) is a set of decision rules for each region $i$:

$$\hat{k}_{i,t+1} = h^k_{it}(\hat{\omega}_{it},z_{it})$$

$$\hat{x}_{it} = h^x_{it}(k_{it},z_{it}),$$

for $t = 0,\ldots,t_2-1$; and a pair of time-invariant decision rules,

$$\hat{k}_{i,t+1} = h^k_i(\hat{\omega}_{it},z_{it})$$

$$\hat{x}_{it} = h^x_i(k_{it},z_{it}),$$

for $t \geq t_2$.





## A6  The endogenous grid point method

This section describes how to the use the "endogenous grid point" method to solve the dynamic program.

### A6.1  The steady-state problem

Start with the steady-state problem (A6), now assuming that $U(c) = \log(c)$:

$$\hat{v}^i(\hat{\omega}_{it}, z_{it}) = \max_{\hat{k}_{i,t+1}} \left[ \log\left( \hat{\omega}_{it} - (1+g^N)(1+g^A)\hat{k}_{i,t+1} \right) + \right. \tag{A8}$$

$$\left. \beta(1+g^N)\,\mathbb{E}_t\left( \hat{v}^i(\hat{\omega}_{i,t+1}, z_{i,t+1}) \right) \right],$$

subject to eq. (A1), (A2), (A4), and the borrowing constraint. In equation (A4), the law of motion for wealth, note that the optimal choice for $\hat{x}_{i,t+1}$ solves:

$$F_x\left( \hat{k}_{i,t+1}^\alpha \left( d_{t+1}^i(T_{i,t+1}) \right)^{1-\alpha}, \hat{x}_{i,t+1} \right) = p_i,$$

where $F_x$ denotes the partial derivative of $F$ with respect to its second argument. Recalling that, in the steady state, $T_{i,t+1} = \overline{T}_i^* + z_{i,t+1}$, this first-order condition implicitly defines a time-invariant decision rule for (scaled) energy use:

$$\hat{x}_{i,t+1} = h_i^x(\hat{k}_{i,t+1}, z_{i,t+1}). \tag{A9}$$

Substitute this decision rule into eq. (A4):

$$\hat{\omega}_{i,t+1} = F\left( \hat{k}_{i,t+1}^\alpha \left( d_{t+1}^i(T_{i,t+1}) \right)^{1-\alpha}, h_i^x(\hat{k}_{i,t+1}, z_{i,t+1}) \right) - p_i h_i^x(\hat{k}_{i,t+1}, z_{i,t+1}) +$$

$$(1-\delta)\hat{k}_{i,t+1},$$

$$\equiv G_i(\hat{k}_{i,t+1}, z_{i,t+1}). \tag{A10}$$

The first-order condition to problem (A8) is then:

$$\beta^{-1}(1+g^A)\left( \hat{\omega}_{it} - (1+g^N)(1+g^A)\hat{k}_{i,t+1} \right)^{-1} = \mathbb{E}_t\left[ \hat{v}_\omega^i(\hat{\omega}_{i,t+1}, z_{i,t+1}) G_i^k(\hat{k}_{i,t+1}, z_{i,t+1}) \right], \tag{A11}$$

where $\hat{v}_\omega^i$ denotes the partial derivative of $\hat{v}^i$ with respect to its first argument and $G_i^k$ denotes the partial derivative of $G_i$ with respect to its first argument. The right-hand side of this first-order condition can be rewritten as:

$$\mathbb{E}_t\left[ \hat{v}_\omega^i(\hat{\omega}_{i,t+1}, z_{i,t+1}) G_i^k(\hat{k}_{i,t+1}, z_{i,t+1}) \right] = \tag{A12}$$

$$\int_{-\infty}^{\infty} \hat{v}_\omega^i(G_i(\hat{k}_{i,t+1}, z_{i,t+1}), z_{i,t+1}) G_i^k(\hat{k}_{i,t+1}, z_{i,t+1}) f(z_{i,t+1}; \rho_i z_{it}, \sigma_i^2)\, dz_{i,t+1} \equiv$$

$$H_i(\hat{k}_{i,t+1}, z_{it}),$$

where $f(z; \mu, \sigma^2)$ is the density function for a random variable $z$ that has a normal distribution with mean $\mu$ and variance $\sigma^2$. So then the first-order condition can be written:

$$\beta^{-1}(1+g^A)\left( \hat{\omega}_{it} - (1+g^N)(1+g^A)\hat{k}_{i,t+1} \right)^{-1} = H_i(\hat{k}_{i,t+1}, z_{it}). \tag{A13}$$





The first step in the endogenous grid point method is to use eq. (A13) to solve for $\hat{\omega}_{it}$ on a grid of points for $\hat{k}_{i,t+1}$ and $z_{it}$, given a guess for $H_i$ on the grid. Let the grid points be $\{\bar{k}'_j\}_{j=1}^{n_k}$ (where $'$ denotes "next period") and $\{\bar{z}_\ell\}_{\ell=1}^{n_\ell}$, respectively, and guess on an initial set of values for $H_i$ on this grid, i.e., $\{\{H_i^0(\bar{k}'_j, \bar{z}_\ell)\}_{\ell=1}^{n_\ell}\}_{j=1}^{n_k}$. Then, for each $(j, \ell)$ pair,

$$\bar{\omega}_{j\ell}^0 = \beta^{-1}(1+g^A)\left(H_i^0(\bar{k}'_j, \bar{z}_\ell)\right)^{-1} + (1+g^N)(1+g^A)\bar{k}'_j$$

thereby determining, for each $\ell$, the optimal choice for savings on the implied grid for $\omega$ (which varies with $\ell$):

$$\bar{k}'_j = h_i^k(\bar{\omega}_{j\ell}^0, \bar{z}_\ell), \quad j = 1, \ldots, n_k. \tag{A14}$$

In the second step, choose a fixed grid for $\omega$, i.e., $\{\bar{\omega}_j\}_{j=1}^{n_\omega}$, where the grid points for $\omega$ in this case do not depend on $z_\ell$. For each $\ell$, interpolate (linearly) using the information in (A14) to determine the optimal choice for savings, $\bar{k}'_{j\ell}$, at the grid points $(\bar{\omega}_j, z_\ell), j = 1 \ldots, n_\omega$:

$$\bar{k}'_{jl} = h_i^k(\bar{\omega}_j, \bar{z}_\ell), \quad j = 1, \ldots, n_\omega. \tag{A15}$$

The information in (A15) can then be used (using bilinear interpolation) to calculate optimal savings choices for any $(\omega, z)$ pair; call this interpolated function $k' = \hat{h}_i^k(\omega, z)$.

In the third step, use the envelope condition to problem (A8) to update the guess on $H_i^0$. In particular, the envelope condition is:

$$\hat{v}_\omega^i(\hat{\omega}_{it}, z_{it}) = \left(\hat{\omega}_{it} - (1+g^N)(1+g^A)\hat{k}_{i,t+1}\right)^{-1} \tag{A16}$$

Update one period:

$$\begin{aligned}
\hat{v}_\omega^i(\hat{\omega}_{i,t+1}, z_{i,t+1}) &= \left(\hat{\omega}_{i,t+1} - (1+g^N)(1+g^A)\hat{k}_{i,t+2}\right)^{-1} \\
&= \left(G_i(\hat{k}_{i,t+1}, z_{i,t+1}) - (1+g^N)(1+g^A)\hat{h}_i^k(\hat{\omega}_{i,t+1}, z_{i,t+1})\right)^{-1} \\
&= \left(G_i(\hat{k}_{i,t+1}, z_{i,t+1}) - (1+g^N)(1+g^A)\hat{h}_i^k(G_i(\hat{k}_{i,t+1}, z_{i,t+1}), z_{i,t+1})\right)^{-1} \\
&\equiv \Lambda_i(\hat{k}_{i,t+1}, z_{i,t+1}).
\end{aligned}$$

To update the guess on $H_i^0$, calculate

$$\begin{aligned}
H_i^1(\hat{k}_{i,t+1}, z_{it}) &= \mathbb{E}_t\left[\Lambda_i(\hat{k}_{i,t+1}, z_{i,t+1})G_i^k(\hat{k}_{i,t+1}, z_{i,t+1})\right] \\
&= \int_{-\infty}^{\infty} \Lambda_i(\hat{k}_{i,t+1}, z_{i,t+1})G_i^k(\hat{k}_{i,t+1}, z_{i,t+1})f(z_{i,t+1}; \rho_i z_{it}, \sigma_i^2)\,dz_{i,t+1} \\
&\approx \pi^{-1/2}\sum_{m=1}^{M} w_m \Lambda_i(\hat{k}_{i,t+1}, z_i^m)G_i^k(\hat{k}_{i,t+1}, z_i^m),
\end{aligned}$$

where $z_i^m = \rho_i z_{it} + \sqrt{2}\sigma_i a_m$ and $(w_m, a_m), m = 1, \ldots, M$, are the Gauss-Hermite weights and abscissas for $M$-point Gaussian quadrature. In particular, calculate $H_i^1(\cdot, \cdot)$ at the grid points $\{\{(\bar{k}'_j, \bar{z}_\ell)\}_{\ell=1}^{n_\ell}\}_{j=1}^{n_k}$, replace $H_i^0$ with $H_i^1$ at these grid points, and then repeat the three steps above until $H_i^0$ and $H_i^1$ are close.




### A6.2 Euler equation errors in the steady-state problem

One way to check the accuracy of the candidate decision rule, $\hat{h}_i^k(\omega, z)$, computed in Section A6.1, to calculate Euler equation errors, expressed in consumption units. These should be exactly zero for any choice of the state variables. To obtain the Euler equation, substitute the envelope condition (A16), updated one period, into the first-order condition (A11):

$$\beta^{-1} G^A \left( \hat{\omega}_{it} - G^{AN} \hat{k}_{i,t+1} \right)^{-1} = \mathbb{E}_t \left[ \left( \hat{\omega}_{i,t+1} - G^{AN} \hat{k}_{i,t+2} \right)^{-1} G_i^k(\hat{k}_{i,t+1}, z_{i,t+1}) \right]. \tag{A17}$$

where $G^A \equiv 1 + g^A$ and $G^{AN} \equiv (1+g^A)(1+g^N)$. Using the candidate decision rule and the function $G_i$, the right-hand side of this equation can be rewritten:

$$\mathbb{E}_t \left[ \left( \hat{\omega}_{i,t+1} - G^{AN} \hat{k}_{i,t+2} \right)^{-1} G_i^k(\hat{k}_{i,t+1}, z_{i,t+1}) \right] =$$
$$\mathbb{E}_t \left[ \left( G_i(\hat{k}_{i,t+1}, z_{i,t+1}) - G^{AN} \hat{h}_i^k(\hat{\omega}_{i,t+1}, z_{i,t+1}) \right)^{-1} G_i^k(\hat{h}_i^k(\hat{\omega}_{it}, z_{it}), z_{i,t+1}) \right] =$$
$$\mathbb{E}_t \left[ \left( G_i(\hat{h}_i^k(\hat{\omega}_{it}, z_{it}), z_{i,t+1}) - G^{AN} \hat{h}_i^k(G_i(\hat{h}_i^k(\hat{\omega}_{it}, z_{it}), z_{i,t+1}), z_{i,t+1}) \right)^{-1} \cdot$$
$$G_i^k(\hat{h}_i^k(\hat{\omega}_{it}, z_{it}), z_{i,t+1}) \right] \equiv$$
$$\Phi_i(\hat{\omega}_{it}, z_{it}),$$

where the conditional expectation in the definition of $\Phi_{it}$ can be approximated via Gaussian quadrature as before. Inverting both sides of (A17) and rearranging, the Euler equation error, $E_i(\hat{\omega}_{it}, z_{it})$, is defined as follows:

$$\begin{aligned} E_i(\hat{\omega}_{it}, z_{it}) &\equiv \hat{\omega}_{it} - G^{AN} \hat{k}_{i,t+1} - \beta^{-1} G^A \, \Phi_i^{-1}(\hat{\omega}_{it}, z_{it}) \\ &= \hat{\omega}_{it} - G^{AN} \hat{h}_i^k(\hat{\omega}_{it}, z_{it}) - \beta^{-1} G^A \, \Phi_i^{-1}(\hat{\omega}_{it}, z_{it}). \end{aligned} \tag{A18}$$

This error should be close to zero for any pair $(\hat{\omega}_{it}, z_{it})$; it would be exactly zero if $\hat{h}_i^k$ were the exact decision rule. The error relative to period-$t$ consumption is:

$$\begin{aligned} \hat{E}_i(\hat{\omega}_{it}, z_{it}) &\equiv \frac{E_i(\hat{\omega}_{it}, z_{it})}{\hat{\omega}_{it} - G^{AN} \hat{k}_{i,t+1}} \\ &= 1 - \frac{\beta^{-1} G^A \, \Phi_i^{-1}(\hat{\omega}_{it}, z_{it})}{\hat{\omega}_{it} - G^{AN} \hat{h}_i^k(\hat{\omega}_{it}, z_{it})}. \end{aligned}$$

### A6.3 The transition problem

Now consider the transition problem:

$$\hat{v}_{it}(\hat{\omega}_{it}, z_{it}) = \max_{\hat{k}_{i,t+1}} \left[ \log \left( \hat{\omega}_{it} - G_{i,t+1}^{AN} d_{it}^{t+1} \hat{k}_{i,t+1} \right) + \beta G_{i,t+1}^N \mathbb{E}_t \left( \hat{v}_{i,t+1}(\hat{\omega}_{i,t+1}, z_{i,t+1}) \right) \right], \tag{A19}$$

where $G_{it}^N \equiv 1 + g_{it}^N$, $G_{it}^A \equiv 1 + g_{it}^A$, $G_{it}^{AN} \equiv (1+g_{it}^N)(1+g_{it}^A)$, and $d_{it}^{t+1} \equiv d_{it}(\overline{T}_{i,t+1})$, subject to eq. (A1), (A2), (A4), and the borrowing constraint. As in Section A6.1, the optimal choice for $\hat{x}_{i,t+1}$ in the law of motion for wealth solves:

$$F_x \left( \hat{k}_{i,t+1}^\alpha \left( d_{t+1}^i(T_{i,t+1}) \right)^{1-\alpha}, \hat{x}_{i,t+1} \right) = p_i,$$




Along the transition path, $T_{i,t+1} = \overline{T}_{i,t+1} + z_{i,t+1}$, so this first-order condition defines a decision rule for (scaled) energy use that now depends on time:

$$\hat{x}_{i,t+1} = h^x_{i,t+1}(\hat{k}_{i,t+1}, z_{i,t+1}). \tag{A20}$$

Substitute this decision rule into (A4):

$$
\begin{aligned}
\quad \hat{\omega}_{i,t+1} &= F\left(\hat{k}^\alpha_{i,t+1}\left(d^i_{t+1}(T_{i,t+1})\right)^{1-\alpha}, h^x_{i,t+1}(\hat{k}_{i,t+1}, z_{i,t+1})\right) - p_i h^x_{i,t+1}(\hat{k}_{i,t+1}, z_{i,t+1}) + \\
&\quad (1-\delta)\hat{k}_{i,t+1}, \\
&\equiv G_{i,t+1}(\hat{k}_{i,t+1}, z_{i,t+1}). \tag{A21}
\end{aligned}
$$

The first-order condition to problem (A19) is then:

$$\beta^{-1} G^A_{i,t+1} d^{t+1}_{it}\left(\hat{\omega}_{it} - G^{AN}_{i,t+1} d^{t+1}_{it} \hat{k}_{i,t+1}\right)^{-1} = \mathbb{E}_t\left[\hat{v}^\omega_{i,t+1}(\hat{\omega}_{i,t+1}, z_{i,t+1}) G^k_{i,t+1}(\hat{k}_{i,t+1}, z_{i,t+1})\right], \tag{A22}$$

where $\hat{v}^\omega_{i,t+1}$ denotes the partial derivative of $\hat{v}_{i,t+1}$ with respect to its first argument, $G^k_{i,t+1}$ denotes the partial derivative of $G_{i,t+1}$ with respect to its first argument, and $G^A_{i,t+1} \equiv 1 + g^A_{i,t+1}$. The right-hand side of this first-order condition can be rewritten as:

$$\mathbb{E}_t\left[\hat{v}^\omega_{i,t+1}(\hat{\omega}_{i,t+1}, z_{i,t+1}) G^k_{i,t+1}(\hat{k}_{i,t+1}, z_{i,t+1})\right] = \tag{A23}$$

$$\int_{-\infty}^{\infty} \hat{v}^\omega_{i,t+1}(G_{i,t+1}(\hat{k}_{i,t+1}, z_{i,t+1}), z_{i,t+1}) G^k_{i,t+1}(\hat{k}_{i,t+1}, z_{i,t+1}) f(z_{i,t+1}; \rho_i z_{it}, \sigma^2_i) dz_{i,t+1} \equiv$$

$$H_{i,t+1}(\hat{k}_{i,t+1}, z_{it}),$$

where $f(z; \mu, \sigma^2)$ is the density function for a random variable $z$ that has a normal distribution with mean $\mu$ and variance $\sigma^2$.
So then the first-order condition can be written:

$$\beta^{-1} G^A_{i,t+1} d^{t+1}_{it}\left(\hat{\omega}_{it} - G^{AN}_{i,t+1} d^{t+1}_{it} \hat{k}_{i,t+1}\right)^{-1} = H_{i,t+1}(\hat{k}_{i,t+1}, z_{it}). \tag{A24}$$

Given knowledge of $H_{i,t+1}(\hat{k}_{i,t+1}, z_{it})$ on a grid points $\{(\bar{k}'_j, \bar{z}_\ell)\}^{n_\ell}_{\ell=1}\}^{n_k}_{j=1}$, the goal is to calculate $H_{it}(\hat{k}_{it}, z_{i,t-1})$ and then continue iterating backwards in a similar fashion to time 0.

To do so, use eq. (A24) to solve for $\hat{\omega}_{it}$ at each grid point:

$$\bar{\omega}_{j\ell} = \beta^{-1} G^A_{i,t+1} d^{t+1}_{it}\left(H_{i,t+1}(\bar{k}'_j, \bar{z}_\ell)\right)^{-1} + G^{AN}_{i,t+1} d^{t+1}_{it} \bar{k}'_j$$

thereby determining, for each $\ell$, the optimal choice for savings, at time $t$, on the implied grid for $\omega$ (which varies with $\ell$):

$$\bar{k}'_j = h^k_{it}(\bar{\omega}_{j\ell}, \bar{z}_\ell), \quad j = 1, \ldots, n_k. \tag{A25}$$





Next, choose a fixed grid for $\omega$, i.e., $\{\bar{\omega}_j\}_{j=1}^{n_\omega}$, where the grid points for $\omega$ in this case do not depend on $z_\ell$. For each $\ell$, interpolate (linearly) using the information in (A25) to determine the optimal choice for savings, $\bar{k}'_{j\ell}$, at the grid points $(\bar{\omega}_j, z_\ell)$, $j = 1 \dots, n_\omega$:

$$\bar{k}'_{jl} = h^k_{it}(\bar{\omega}_j, \bar{z}_\ell), \quad j = 1, \dots, n_\omega. \tag{A26}$$

The information in (A26) can then be used (using bilinear interpolation) to calculate optimal savings choices for any $(\omega, z)$ pair; call this interpolated function $k' = \hat{h}^k_{it}(\omega, z)$.

The envelope condition for the transition problem (A19) is:

$$
\begin{aligned}
v^\omega_{it}(\hat{\omega}_{it}, z_{it}) &= \left(\hat{\omega}_{it} - G^{AN}_{i,t+1} d^{t+1}_{it} \hat{k}_{i,t+1}\right)^{-1} \\
&= \left(G_{it}(\hat{k}_{it}, z_{it}) - G^{AN}_{i,t+1} d^{t+1}_{it} \hat{h}^k_{it}(\hat{\omega}_{it}, z_{it})\right)^{-1} \\
&= \left(G_{it}(\hat{k}_{it}, z_{it}) - G^{AN}_{i,t+1} d^{t+1}_{it} \hat{h}^k_{it}(G_{it}(\hat{k}_{it}, z_{it}), z_{it})\right)^{-1}
\end{aligned}
$$


By definition,

$$
\begin{aligned}
H_{it}(\hat{k}_{it}, z_{i,t-1}) &= \mathbb{E}_{t-1}\left[\hat{v}^\omega_{it}(\hat{\omega}_{it}, z_{it}) G^k_{it}(\hat{k}_{it}, z_{it})\right] \\
&= \mathbb{E}_{t-1}\left[\left(G_{it}(\hat{k}_{it}, z_{it}) - G^{AN}_{i,t+1} \hat{h}^k_{it}(G_{it}(\hat{k}_{it}, z_{it}), z_{it})\right)^{-1} G^k_{it}(\hat{k}_{it}, z_{it})\right].
\end{aligned}
$$

Use this equation, together with the already-calculated approximate decision $\hat{h}^k_{it}$, to calculate $\{\{H_{it}(\bar{k}'_j, \bar{z}_\ell)\}_{\ell=1}^{n_\ell}\}_{j=1}^{n_k}$ and then iterate backwards following the same steps as laid out above.

For $t \geq t_2$, $g^A_{it} = g^A$, $g^N_{it} = g^N$, and $\overline{T}_{it} = \overline{T}^*_i$ as in the steady-state problem. Therefore, when $t = t_2 - 1$, the first-order condition (A24) is:

$$\beta^{-1} G^A d^{t+1}_{it} \left(\hat{\omega}_{it} - G^{AN} d^{t+1}_{it} \hat{k}_{i,t+1}\right)^{-1} = H_i(\hat{k}_{i,t+1}, z_{it}),$$

where $H_i$ has already been computed when solving the steady-state problem. $H_i$ is, therefore, the starting point for the backwards iterations.

### A6.4    Euler equation errors in the transition problem

By analogy to the Euler eq. (A17) in the steady-state problem, the Euler equation along a transition is:

$$
\begin{aligned}
\beta^{-1} G^A_{i,t+1} d^{t+1}_{it} \left(\hat{\omega}_{it} - G^{AN}_{i,t+1} d^{t+1}_{it} \hat{k}_{i,t+1}\right)^{-1} &= \\
\mathbb{E}_t\left[\left(\hat{\omega}_{i,t+1} - G^{AN}_{i,t+2} d^{t+2}_{i,t+1} \hat{k}_{i,t+2}\right)^{-1} G^k_{i,t+1}(\hat{k}_{i,t+1}, z_{i,t+1})\right]. &
\end{aligned}
\tag{A27}
$$





The right-hand side of this equation can be rewritten:

$$
\mathbb{E}_t\left[\left(\hat{\omega}_{i,t+1} - G_{i,t+2}^{AN}\, d_{i,t+1}^{t+2}\, \hat{k}_{i,t+2}\right)^{-1} G_i^k(\hat{k}_{i,t+1}, z_{i,t+1})\right] =
$$

$$
\mathbb{E}_t\left[\left(G_{i,t+1}(\hat{k}_{i,t+1}, z_{i,t+1}) - G_{i,t+2}^{AN}\, d_{i,t+1}^{t+2}\, \hat{h}_{i,t+1}^k(\hat{\omega}_{i,t+1}, z_{i,t+1})\right)^{-1} G_{i,t+1}^k(\hat{h}_{it}^k(\hat{\omega}_{it}, z_{it}), z_{i,t+1})\right] =
$$

$$
\mathbb{E}_t\left[\left(G_{i,t+1}(\hat{h}_{it}^k(\hat{\omega}_{it}, z_{it}), z_{i,t+1}) - G_{i,t+2}^{AN}\, d_{i,t+1}^{t+2}\, \hat{h}_{i,t+1}^k(G_{i,t+1}(\hat{k}_{i,t+1}, z_{i,t+1}), z_{i,t+1})\right)^{-1}\right. \cdot
$$

$$
\left. G_{i,t+1}^k(\hat{h}_{it}^k(\hat{\omega}_{it}, z_{it}), z_{i,t+1})\right] =
$$

$$
\mathbb{E}_t\left[\left(G_{i,t+1}(\hat{h}_{it}^k(\hat{\omega}_{it}, z_{it}), z_{i,t+1}) - G_{i,t+2}^{AN}\, d_{i,t+1}^{t+2}\, \hat{h}_{i,t+1}^k(G_{i,t+1}(\hat{h}_{it}^k(\hat{\omega}_t, z_{it}), z_{i,t+1}), z_{i,t+1})\right)^{-1}\right. \cdot
$$

$$
\left. G_{i,t+1}^k(\hat{h}_{it}^k(\hat{\omega}_{it}, z_{it}), z_{i,t+1})\right] \equiv
$$

$$
\Phi_{it}(\hat{\omega}_{it}, z_{it}),
$$

where the conditional expectation in the definition of $\Phi_{it}$ can be approximated via Gaussian quadrature. (Note that, in the last period of the transition, when $t = t_2 - 1$, $\hat{h}_{i,t+1}^k = \hat{h}_i^k$, i.e., the decision rule from the steady-state problem.)

Inverting both sides of (A27) and rearranging, the Euler equation error, $E_{it}(\hat{\omega}_{it}, z_{it})$, at time $t$ is:

$$
\begin{aligned}
E_{it}(\hat{\omega}_{it}, z_{it}) &\equiv \hat{\omega}_{it} - G_{i,t+1}^{AN}\, d_{it}^{t+1}\, \hat{k}_{i,t+1} - \beta^{-1} G_{i,t+1}^A\, d_{it}^{t+1}\, \Phi_{it}^{-1}(\hat{\omega}_{it}, z_{it}) \\
&= \hat{\omega}_{it} - G_{i,t+1}^{AN}\, d_{it}^{t+1}\, \hat{h}_{it}^k(\hat{\omega}_{it}, z_{it}) - \beta^{-1} G_{i,t+1}^A\, d_{it}^{t+1}\, \Phi_{it}^{-1}(\hat{\omega}_{it}, z_{it}).
\end{aligned} \tag{A28}
$$

This error should be close to zero for any pair $(\hat{\omega}_{it}, z_{it})$. The error relative to period-$t$ consumption is:

$$
\begin{aligned}
\hat{E}_{it}(\hat{\omega}_{it}, z_{it}) &\equiv \frac{E_{it}(\hat{\omega}_{it}, z_{it})}{\hat{\omega}_{it} - G_{i,t+1}^{AN}\, d_{it}^{t+1}\, \hat{k}_{i,t+1}} \\
&= 1 - \frac{\beta^{-1} G_{i,t+1}^A\, d_{it}^{t+1}\, \Phi_{it}^{-1}(\hat{\omega}_{it}, z_{it})}{\hat{\omega}_{it} - G_{i,t+1}^{AN}\, d_{it}^{t+1}\, \hat{h}_{it}^k(\hat{\omega}_{it}, z_{it})}.
\end{aligned}
$$

## A7 Simulating forwards using NorESM2

This section describes how to simulate the model using NorESM2 up to, and including, year $t_0 < t_1$.

Each year $t$ is divided into $J$ subperiods indexed by $j$. Let $T_{itj}$ be temperature in region $i$ in subperiod $j$ of year $t$ and define average temperature in region $i$ in year $t$ as follows:

$$
T_{it} = J^{-1} \sum_{j=1}^{J} T_{itj}.
$$

Define $a_{it} \equiv A_{it} D(\overline{T}_{it})$. The initial value of $a_{i0}$ in each region is chosen to match regional data in real-world year $R$. The ratio of successive values of $a_{it}$ is given by:

$$
\begin{aligned}
\frac{a_{it}}{a_{i,t-1}} &= \frac{A_{it}}{A_{i,t-1}} \frac{D(\overline{T}_{it})}{D(\overline{T}_{i,t-1})} \\
&= \left(1 + g_{it}^A\right) d_{t-1}^i(\overline{T}_{it}),
\end{aligned}
$$





so that the sequence $\{a_{it}\}$, for $t \geq 1$, obeys the recursion

$$a_{it} = \left(1 + g_{it}^A\right) d_{t-1}^i(\overline{T}_{it}) \, a_{i,t-1}, \tag{A29}$$

where the sequence $\{\overline{T}_{it}\}$ depends on the sequence $\{S_t\}$ taken as given when solving each region's dynamic programming problem.

At the beginning of year 0, $\hat{k}_{i0}$ and $\hat{x}_{i0}$ are chosen in each region to match real-world data in year $R$. The value of cumulative
emissions in year 0, $S_0$, is also chosen to match real-world data in year $R$.

The simulation in any given year $t$ proceeds in the following steps, starting with $t = 0$:

1. If $t \geq 1$, calculate $a_{it}$ according to (A29). (Recall that $a_{i0}$ is set to match regional data at time 0, i.e., in real-world year $R$.)

2. If $t = 0$, calculate regional emissions according to: $e_{i0} = \phi_{i0} N_{i0} a_{i0} \hat{x}_{i0}$, $i = 1, \ldots, M$. If $t \geq 1$, use expected energy use, $\mathbb{E}_{t-1}[\hat{x}_{it}]$, as calculated in the simulation for year $t - 1$, to calculate (expected) emissions in each region $i$:

   $$e_{it} = \phi_{it} N_{it} a_{it} \, \mathbb{E}_{t-1}[\hat{x}_{it}] + e_{i,t-1}^z - e_{i,t-1},$$

   where $e_{t-1}^z$ is actual emissions in region $i$ in period $t - 1$, as defined in step 7 below. Expected emissions are used to
drive NorESM2 in step 3. The term $e_{i,t-1}^z - e_{i,t-1}$ corrects for the deviation between actual and expected emissions in any given period by adding this correction to expected emissions in the subsequent period.

3. Use $\{e_{it}\}_{i=1}^M$ in NorESM2 to generate time paths for temperature in each region in each subperiod of year $t$, i.e., $\{\{T_{itj}\}_{j=1}^J\}_{i=1}^M$, and use these in turn to calculate average temperature in region $i$ in year $t$, i.e.,

   $$T_{it} = J^{-1} \sum_{j=1}^J T_{itj}, \quad i = 1, \ldots, M.$$

   For each region, calculate the deviation of its temperature in year $t$ from its expected temperature in year $t$, i.e., $z_{it} = T_{it} - \overline{T}_{it}$.

4. Calculate actual scaled energy use in each region: $\hat{x}_{it}^z = h_{it}^x(\hat{k}_{it}, z_{it})$, $i = 1, \ldots, M$. Note that the function $\hat{h}_{it}^x$, as defined
in (A20), has a closed-form expression if the production function, $F$, has a particular functional form. (Recall too that the function $h_{it}^x$ depends on $\overline{T}_{it}$.)

5. Insert $\hat{x}_{it}^z$ into (A4) to calculate (scaled) wealth in each region $i$ in year $t$:

   $$\hat{\omega}_{it} = F\left(\hat{k}_{it}^\alpha \left(d_t^i(T_{it})\right)^{1-\alpha}, \hat{x}_{it}^z\right) - p_i \hat{x}_{it}^z + (1 - \delta)\hat{k}_{it}, \quad i = 1, \ldots, M.$$

   (Recall too that $d_t^i(T_{it})$ depends on $\overline{T}_{it}$.)

6. Use the year-$t$ decision rules to calculate regional savings in year t: $\hat{k}_{i,t+1} = h_{it}^k(\hat{\omega}_{it}, z_{it})$, $i = 1, \ldots, M$. The function $h_{it}^k$
is known only on a two-dimensional grid of points, so use bilinear interpolation to calculate it off the grid, as explained in Section **??**.



7. Calculate actual emissions in each region: $e^z_{it} = \phi_{it} N_{it} a_{it} \hat{x}^z_{it}$, $i = 1, \dots, M$.

8. Calculate the expected value of energy use in each region in period $t + 1$:

$$\mathbb{E}_t[\hat{x}_{i,t+1}] = \mathbb{E}_t\left[\hat{h}^x_{i,t+1}(\hat{k}_{i,t+1}, z_{i,t+1})\right].$$

Approximate the conditional expectation in this expression by replacing $z_{i,t+1}$ with its conditional expectation $\rho_i z_{it}$:
$\mathbb{E}_t[\hat{x}_{i,t+1}] = \hat{h}^x_{i,t+1}(\hat{k}_{i,t+1}, \rho_i z_{it})$.

9. The simulation in year $t$ is now complete: return to step 1 to conduct the simulation for year $t + 1$.

Having completed the entire simulation, calculate actual cumulative global emissions at the beginning of each year according
to: $S^z_{t+1} = S^z_t + E^z_t$, where $E^z_t$ is actual aggregate emissions in period $t$:

$$E^z_t = \sum_{i=1}^{M} e^z_{it}.$$

(When $t = 0$, set $S^z_t = S_0$.) Then calculate a new sequence for expected regional temperature using the sequence $\{S^z_t\}$: $\overline{T}^z_{it} = \overline{T}_i + \gamma_{i1} S^z_t + \gamma_{i2} (S^z_t)^2$ for $t \geq 0$. Finally, for each $t$, calculate the differences $S^z_t - S_t$ and $\{\overline{T}^z_{it} - \overline{T}_{it}\}_{i=1}^{M}$ and confirm that
these differences are small.

*Author contributions.* JB led the coupling process, performed the simulation involving NorESM2 (including the coupled simulations), produced many of the figures, and led the writing of the paper with help from the co-authors. AAS contributed to the coupling and help write
the paper, especially the economics. HC contributed to the coupling, performed the standalone DIAM simulations, and produced the map
figures. TS (together with AAS) conceived the idea of coupling the two models, contributed to the coupling, and helped write the paper.

*Competing interests.* The authors declare that they have no conflict of interest.

*Acknowledgements.* This research was funded by the Research Council of Norway through research grant number 281071 ("Climate Change
Modeling and Prediction of Economic Impact") and grant number 309377 ("Integrating Macroeconomics, Climate Physics and Game Theory
for Innovative Education and Research"). The NorESM2 simulations were performed on resources provided by Sigma2 - the National
Infrastructure for High-Performance Computing and Data Storage in Norway through computing and storage grants number nn9600k and
ns9600k, respectively. We would like to acknowledge the Planetary Solutions Project at Yale University and the Cowles Foundation for
Research in Economics for financial support.





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
