# Peer review of "NorESM2–DIAM: A coupled model for investigating global and regional climate-economy interactions"

_EGUsphere, 2025_

## Referee Comment (RC1)

Referee Report on "NorESM2-DIAM: A coupled model for investigating global and regional climate-economy interactions"

by Jenny Bjordal, Anthony A. Smith Jr., Henri Cornec, and Trude Storelymo

**1 A brief summary of the paper**

The paper by Bjordal, Smith, Cornec, and Storelvmo introduces **NorESM2**–**DIAM**, a coupled framework linking a state-of-the-art Earth System Model (NorESM2) with a high-resolution Integrated Assessment Model (DIAM). The objective is to study global and regional climate–economy interactions with a level of spatial and physical detail unavailable in conventional IAMs. The economic model covers about 19,000 regions worldwide, where agents optimize consumption, investment, and energy use given temperature-dependent productivity, while NorESM2 simulates the climate response to their emissions.

The coupling is achieved through an iterative fixed-point procedure: DIAM generates emissions that feed into NorESM2, which in turn returns regional temperature and weather outcomes that affect productivity. A simplified statistical version of DIAM ensures tractability and internal consistency of expectations. The model reproduces global emission and temperature trajectories consistent with the NorESM2 climate response and produces credible spatial patterns of warming and economic change.

Quantitatively, the global mean temperature is projected to rise by about 3.5°C by 2100, leading to a decline of roughly 35% in global GDP per capita relative to trend. Around two-thirds of this decline stems from demographic shifts toward warmer, poorer regions, and about one-quarter reflects direct climate damages. Regional outcomes are highly heterogeneous: colder regions gain modestly from warming, while tropical and subtropical regions suffer substantial losses. The framework also shows that internal climate variability generates nontrivial fluctuations in GDP, highlighting the importance of weather variability for economic outcomes.

**2 Assessment**

This is an exciting and important methodological paper. To the best of my knowledge, it is the first study that succeeds in coupling a spatially highly resolved economic model with a fully fledged Earth System Model (ESM). The authors make a major step toward moving beyond externally imposed RCP or SSP scenarios, into a framework in which emissions are endogenously generated by economic agents. I applied the authors for this ambitious and timely contribution.

The paper is clearly written, conceptually transparent, and the authors provide an honest and thorough discussion of the methodological challenges and current limitations. It is in excellent shape and, in my view, fully merits publication.

I have only a few minor stylistic suggestions, none of which should delay publication.

**3 Minor Comments**

1. It might be helpful for readers if the authors included a schematic diagram (e.g., in Section 2) illustrating the structure of the IAM in addition to its algebraic and textual description. Such a figure could clarify where exactly the coupling ("handshake") between NorESM2 and the economic model occurs.

- 2. The quadratic functional form of temperature in Equation (1) is introduced on page 7, but its justification appears only later in Section 4.7. Since many readers may be more familiar with the linear specification, it could be useful either to move the discussion in Section 4.7 closer to the equation's introduction or to insert a brief forward reference to where the justification is provided.
- 3. Appendix A.6.2 explains how the numerical accuracy of the procedure can be assessed, but I could not find a summary table reporting the associated error statistics. Including such a table would help the reader gauge the quantitative reliability of the solution method.
- 4. On page 40, line 880, the text reads "as explained in Section??". It seems that a LATEX cross-reference failed to compile correctly and should be fixed.

---

## Referee Comment (RC2)

**Review** of manuscript "NorESM2–DIAM: A coupled model for investigating global and regional climate-economy interactions" by Jenny Bjordal, Anthony A. Smith, Jr., Henri Cornec, and Trude Storelymo

Bjordal et al., 2025 have coupled an Earth System Model, NorESM2, with an Integrated Assessment Model, DIAM, to create a new coupled model, NorESM2-DIAM. The two models are coupled in a continuous fashion and exchange information bidirectionally. Information exchange between the two models occurs at an annual time scale and a regional level and includes annual carbon emissions being passed from DIAM to NorESM2 and temperature values passed from NorESM2 to DIAM. The coupled model is used to study the economic impact of climate change. The manuscript is well written, and the development of this new coupled model addresses the critical need for models that incorporate human-Earth feedbacks. The manuscript should be published after addressing the following comments.

The authors have mentioned, "To our knowledge, it is the first framework to fully couple an ESM with a high-resolution IAM". This is incorrect. The Energy Exascale Earth System Model (E3SM) was coupled with GCAM, an Integrated Assessment Model, in a bidirectional synchronous fashion a few years ago (Thornton et al., 2017; Calvin et al., 2019), and earlier this year, a manuscript documenting a newer version of the coupled E3SM-GCAM model was published (Di Vittorio et al., 2025). These manuscripts should be cited, and comparison should be performed of the similarities and differences in the coupling framework and the impact of two-way coupling on the human and Earth system using the two different coupled models.

**Minor Comment:**

Addition of a schematic figure that shows the coupling between NorESM2 and DIAM will enhance the readability of the manuscript.

**References:**

- Thornton, P. E., Calvin, K., Jones, A. D., Di Vittorio, A. V., Bond-Lamberty, B., Chini, L., et al. (2017). Biospheric feedback effects in a synchronously coupled model of human and Earth systems. Nature Climate Change, 7, 496–500. <a href="https://doi.org/10.1038/nclimate3310">https://doi.org/10.1038/nclimate3310</a>
- Calvin, K., Bond-Lamberty, B., Jones, A., Shi, X., Di Vittorio, A., & Thornton, P. (2019). Characteristics of human-climate feedbacks differ at different radiative forcing levels. Global and Planetary Change, 180, 126–135. https://doi.org/10.1016/j.gloplacha.2019.06.003
- Di Vittorio, A. V., Sinha, E., Hao, D., Singh, B., Calvin, K. V., Shippert, T., et al. (2025). E3SM-GCAM: A synchronously coupled human component in the E3SM Earth system model enables novel human-Earth feedback research. Journal of Advances in Modeling Earth Systems, 17, e2024MS004806.
  https://doi.org/10.1029/2024MS004806

---

## Author Comment (AC1)

**Response to referee comments**

Referee comments are in black.

Responses are in blue. We indicate where in the text the changes can be found by L followed by a number that gives the line number.

*Manuscript citations are in italic with changes in red.*

**1 A brief summary of the paper:**

The paper by Bjordal, Smith, Cornec, and Storelvmo introduces NorESM2–DIAM, a coupled framework linking a state-of-the-art Earth System Model (NorESM2) with a high-resolution Integrated Assessment Model (DIAM). The objective is to study global and regional climate–economy interactions with a level of spatial and physical detail unavailable in conventional IAMs. The economic model covers about 19,000 regions worldwide, where agents optimize consumption, investment, and energy use given temperature-dependent productivity, while NorESM2 simulates the climate response to their emissions. The coupling is achieved through an iterative fixed-point procedure: DIAM generates emissions that feed into NorESM2, which in turn returns regional temperature and weather outcomes that affect productivity. A simplified statistical version of DIAM ensures tractability and internal consistency of expectations. The model reproduces global emission and temperature trajectories consistent with the NorESM2 climate response and produces credible spatial patterns of warming and economic change. Quantitatively, the global mean temperature is projected to rise by about $3.5°C$ by 2100, leading to a decline of roughly 35% in global GDP per capita relative to trend. Around two-thirds of this decline stems from demographic shifts toward warmer, poorer regions, and about one-quarter reflects direct climate damages. Regional outcomes are highly heterogeneous: colder regions gain modestly from warming, while tropical and subtropical regions suffer substantial losses. The framework also shows that internal climate variability generates nontrivial fluctuations in GDP, highlighting the importance of weather variability for economic outcomes.

**2 Assessment**

This is an exciting and important methodological paper. To the best of my knowledge, it is the first study that succeeds in coupling a spatially highly resolved economic model with a fully fledged Earth System Model (ESM). The authors make a major step toward moving beyond externally imposed RCP or SSP scenarios, into a framework in which emissions are endogenously generated by economic agents. I applaud the authors for this ambitious and timely contribution. The paper is clearly written, conceptually transparent, and the authors provide an honest and thorough discussion of the methodological challenges and current limitations. It is in excellent shape and, in my view, fully merits publication.

I have only a few minor stylistic suggestions, none of which should delay publication.

Thank you for the nice summary of our paper, as well as for the encouraging overall assessment. We are grateful for the thorough reading of the paper and the helpful suggestions.

**3 Minor Comments**

1. It might be helpful for readers if the authors included a schematic diagram (e.g., in Section 2) illustrating the structure of the IAM in addition to its algebraic and textual description. Such a figure could clarify where exactly the coupling ("handshake") between NorESM2 and the economic model occurs.

A good suggestion. We have added such a schematic in section 2.

L101: *The two components of NorESM2–DIAM are coupled via a continuous, bidirectional flow of information* *as illustrated in Fig. 1.*

[Figure]

***Figure 1****. Schematic overview of the NorESM2–DIAM coupling and internal interactions. NorESM2 provides regional one-year-mean temperatures for the current model year to DIAM (dotted arrows indicate exchange between models). Regional temperature directly affects regional GDP, which in turn determines regional wealth (solid arrows indicate exchanges happening within one model for the current one-year time step of the coupled model). Based on regional temperature and wealth, each region then makes decisions about savings and energy use using pre-computed decision rules derived from the standalone version of DIAM. Within DIAM, savings affects GDP in the next model year (dotted arrows indicate exchanges happening within one model in the next time step of the coupled model). Energy use determines next year's emissions, which are provided to NorESM2. Finally, to complete the cycle, the different modules of NorESM2 interact to generate*

*new regional temperatures. Note that the modules interact through a coupler, and the timing varies between modules, but they all exchange information at least once every 24 hours (Seland et al., 2020; Danabasoglu et al., 2020).*

2. The quadratic functional form of temperature in Equation (1) is introduced on page 7, but its justification appears only later in Section 4.7. Since many readers may be more familiar with the linear specification, it could be useful either to move the discussion in Section 4.7 closer to the equation's introduction or to insert a brief forward reference to where the justification is provided.

We have changed the forward reference from Section 4 to Section 4.7 for easier navigation in the text, and also made it clearer that the explanation will be found in this section. (L207-208)

*The justification for using a quadratic rather than linear functional form , along with details on how the parameters $\gamma_{i1}$ and $\gamma_{i2}$ are estimated from data, is provided  in Sect. 4.7.*

3. Appendix A.6.2 explains how the numerical accuracy of the procedure can be assessed, but I could not find a summary table reporting the associated error statistics. Including such a table would help the reader gauge the quantitative reliability of the solution method.

A short section summarising the errors has been added. (L886-892)

***A6.5 Euler equation errors: numerical values***
*To check the accuracy of the computed decision rules, both along the transition and in the steady state, we calculated Euler equations errors as described in Sect. A6.2 and A6.4. These errors are very close to zero. In particular, the average error relative to consumption (averaging across both regions and the values of the state variables) is less than 0.0024% (in absolute value) in every time period. In addition, the average of the absolute values of the errors is less than 0.021% in very time period. Finally, apart from a very small number of outliers (fewer than 10), the largest error—looking across all regions, time periods, and states— is smaller than 0.6% in absolute value.*

4. On page 40, line 880, the text reads "as explained in Section ??". It seems that a LATEX cross-reference failed to compile correctly and should be fixed.

Thank you for catching this. We have removed this cross-reference, as it should not be there. (L917)

*The function $h^k_{it}$ is known only on a two-dimensional grid of points, so use bilinear interpolation to calculate it off the grid.*

---

## Author Comment (AC2)

**Response to referee comments**

Referee comments are in black.
Responses are in blue. We indicate where in the text the changes can be found by L followed by a number that gives the line number.
*Manuscript citations are in italic with changes in red.*

Bjordal et al., 2025 have coupled an Earth System Model, NorESM2, with an Integrated Assessment Model, DIAM, to create a new coupled model, NorESM2-DIAM. The two models are coupled in a continuous fashion and exchange information bidirectionally. Information exchange between the two models occurs at an annual time scale and a regional level and includes annual carbon emissions being passed from DIAM to NorESM2 and temperature values passed from NorESM2 to DIAM. The coupled model is used to study the economic impact of climate change. The manuscript is well written, and the development of this new coupled model addresses the critical need for models that incorporate human-Earth feedbacks. The manuscript should be published after addressing the following comments.

Thank you for the clear summary and for the positive evaluation of our contribution. We appreciate the careful review and helpful comments.

1. The authors have mentioned, "To our knowledge, it is the first framework to fully couple an ESM with a high-resolution IAM". This is incorrect. The Energy Exascale Earth System Model (E3SM) was coupled with GCAM, an Integrated Assessment Model, in a bidirectional synchronous fashion a few years ago (Thornton et al., 2017; Calvin et al., 2019), and earlier this year, a manuscript documenting a newer version of the coupled E3SM-GCAM model was published (Di Vittorio et al., 2025). These manuscripts should be cited, and comparison should be performed of the similarities and differences in the coupling framework and the impact of two-way coupling on the human and Earth system using the two different coupled models.

Thank you for pointing out these models to us. We still believe this is the first coupling with a high-resolution cost-benefit model, and have adjusted the abstract accordingly. (L9)

*To our knowledge, it is the first framework to fully couple an ESM with a high-resolution cost-benefit IAM.*

Additionally, as you suggest, we have added a couple of paragraphs discussing the similarities and differences between the coupling framework of our model and similar coupled models like E3SM-GCAM. (This also included some small adjustments to the previous paragraphs for better flow.) (L57-89)

*… Finally, NorESM2--DIAM is a cost-benefit IAM : economic agents (consumers and firms) in the model solve explicitly-specified dynamic decision problems with well-defined objectives. It can therefore provide quantitative assessments of the welfare effects  of a wide range of scenarios for climate policy---from laissez-faire to optimal carbon taxation---both across time and space.*

*The primary goal of this paper, however, is to demonstrate, using a prototype version of NorESM2--DIAM, how to tackle two key methodological challenges in coupling an ESM with a dynamic, high-resolution economic model grounded in dynamic optimization. First, the two models operate on vastly different time scales. Second, the economic model incorporates forward-looking behavior: the decisions of agents  depend on their expectations about the future behavior of the climate, which is itself influenced by those very decisions. Achieving consistency between agents' expectations and the climate trajectory thus requires solving for an interdependent equilibrium.*

*Successfully addressing these challenges lays the groundwork for using NorESM2–DIAM as a platform to explore the spatial and temporal dimensions of climate–economy interactions, and to assess climate policy with a degree of geophysical and economic realism that is rare in existing IAMs. This platform contributes to a small but growing literature using dynamic, forward-looking, structural economic models to study the spatial effects of climate change (see, for example, Brock et al., 2014; Desmet and Rossi-Hansberg, 2015; Fried, 2022; Krusell and Smith, 2022; Rudik et al., 2021; Bilal and Rossi-Hansberg, 2023; Cruz and Rossi-Hansberg, 2024; Kubler, 2023; Kotlikoff et al., 2024).*

*Our approach to coupling an ESM and an IAM, embodied in NorESM2–DIAM, contrasts with the approach taken in iESM (Collins et al., 2015; Thornton et al., 2017; Calvin and Bond-Lamberty, 2018) and E3SM–GCAM (Di Vittorio et al., 2025), two other frameworks that couple an ESM and an IAM. The main difference is that both iESM and E3SM–GCAM couple an ESM with the Global Change Assessment Model (GCAM), a process-based rather than a cost-benefit IAM. Although both DIAM and GCAM are dynamic, recursive models, in DIAM agents make decisions taking into account the entire future time horizon, whereas GCAM solves for outcomes one step at a time, considering only the current state.*

*The two approaches also differ in spatial resolution and sectoral detail. GCAM represents multiple sectors—including energy, industry, transport, agriculture, and land use—but divides the world into only 14 (iESM) or 32 (E3SM-GCAM)*

*socioeconomic regions. In contrast, NorESM2–DIAM contains only a single sector, focusing directly on gross domestic product (GDP), but at a very high degree of spatial resolution (1°×1° cells), enabling high-resolution analysis of the impacts of climate and weather on GDP and emissions.*

*Finally, the three models differ in how they represent climate–economy interactions. iESM and E3SM-GCAM exchange biogeochemical variables from the ESM to GCAM, whereas in our framework, temperature directly affects the economy through the productivity of labor. GCAM also explicitly represents agriculture and land use, allowing iESM and E3SM–GCAM to generate land-mediated feedbacks that are absent in NorESM–DIAM. Thus, although all three frameworks couple an IAM with an ESM, our approach employs a fundamentally different IAM, providing a complementary perspective to the two existing frameworks.*

**Minor comment:**

2. Addition of a schematic figure that shows the coupling between NorESM2 and DIAM will enhance the readability of the manuscript.

A good suggestion. We have added such a schematic in section 2.

L101: *The two components of NorESM2–DIAM are coupled via a continuous, bidirectional flow of information as illustrated in Fig. 1.*

[Figure]

**Figure 1**. *Schematic overview of the NorESM2–DIAM coupling and internal interactions. NorESM2 provides regional one-year-mean temperatures for the current model year to DIAM (dotted arrows indicate exchange between models). Regional temperature directly affects regional GDP, which in turn determines regional wealth (solid arrows indicate exchanges happening within one model for the*

*current one-year time step of the coupled model). Based on regional temperature and wealth, each region then makes decisions about savings and energy use using pre-computed decision rules derived from the standalone version of DIAM. Within DIAM, savings affects GDP in the next model year (dotted arrows indicate exchanges happening within one model in the next time step of the coupled model). Energy use determines next year's emissions, which are provided to NorESM2. Finally, to complete the cycle, the different modules of NorESM2 interact to generate new regional temperatures. Note that the modules interact through a coupler, and the timing varies between modules, but they all exchange information at least once every 24 hours (Seland et al., 2020; Danabasoglu et al., 2020).*

---

## Author Comment (AC3)

**Response to referee comments**

Referee comments are in black.

Responses are in blue. We indicate where in the text the changes can be found by L followed by a number that gives the line number.

*Manuscript citations are in italic with changes in red.*

This study presents a new framework that couples an Earth System Model with a spatially disaggregated Integrated Assessment Model to examine how climate change (i.e., temperature changes) interacts with the economy. The baseline simulation in the study shows that the economic impacts of global warming differ substantially across regions and that internal climate variability leads to significant volatility in regional GDP, emphasizing the value of high-resolution economic impact assessments. The model that this study presents fills an important gap left by previous frameworks that used coarse spatial aggregation, simplified climate representation, or weak coupling between climate and the economy. However, several aspects should be addressed before the manuscript is ready for publication.

Thank you for the careful reading and constructive comments, which we will address in our revision.

**Major Comments:**

1. There are several existing coupled ESM-IAM frameworks (e.g., E3SM–GCAM (Di Vittorio et al., 2025), iESM (Collins et al., 2015; Thornton et al., 2017)), and it would be helpful for the authors to explicitly acknowledge these efforts and situate NorESM2–DIAM within this broader context. Unlike E3SM–GCAM and iESM, which exchange $CO_2$ emissions, terrestrial productivity, and land-use information, NorESM2–DIAM currently exchanges only $CO_2$ emissions and temperature, with economic impacts represented through an aggregate productivity function.

   While this design allows for fine spatial resolution and transparent temperature–productivity relationships, it omits key land-mediated feedbacks, e.g., those related to land cover, albedo, soil carbon, and evapotranspiration,that are represented in NorESM2's land module. I understand that as a spatially disaggregated macroeconomic IAM, DIAM occupies a distinct niche relative to process-based IAMs like GCAM, IMAGE, and MESSAGE, which explicitly simulate land, energy, and technological dynamics. A concise discussion of these trade-offs, what NorESM2–DIAM gains in spatial detail and simplicity, and what it sacrifices in process feedbacks, would help readers clearly understand its comparative advantages

and intended applications.

references:

Di Vittorio, A. V., Sinha, E., Hao, D., Singh, B., Calvin, K. V., Shippert, T., ... & Bond‑Lamberty, B. (2025). E3SM‑GCAM: A synchronously coupled human component in the E3SM Earth system model enables novel human‑Earth feedback research. Journal of Advances in Modeling Earth Systems, 17(6), e2024MS004806.

Collins, William D., Anthony P. Craig, John E. Truesdale, A. V. Di Vittorio, Andrew D. Jones, Benjamin Bond-Lamberty, Katherine V. Calvin et al. "The integrated Earth system model version 1: formulation and functionality." Geoscientific Model Development 8, no. 7 (2015): 2203-2219.

Thornton, P. E., Calvin, K., Jones, A. D., Di Vittorio, A. V., Bond-Lamberty, B., Chini, L., ... & Hurtt, G. (2017). Biospheric feedback effects in a synchronously coupled model of human and Earth systems. Nature Climate Change, 7(7), 496-500.

Thank you for pointing out these models. We agree that this should be included and have added a couple of paragraphs that compare our model with models like iESM and E2SM-GCAM. (This also included some small adjustments to the previous paragraphs for better flow.) (L57-89)

*… Finally, NorESM2--DIAM is a cost-benefit IAM* *: economic agents (consumers and firms) in the model solve explicitly-specified dynamic decision problems with well-defined objectives.  It can therefore provide quantitative assessments of the welfare effects*  *of a wide range of scenarios for climate policy---from laissez-faire to optimal carbon taxation---both across time and space.*

*The primary goal of this paper, however, is to demonstrate, using a prototype version of NorESM2--DIAM, how to tackle two key methodological challenges in coupling an ESM with a dynamic, high-resolution economic model grounded in dynamic optimization. First, the two models operate on vastly different time scales. Second, the economic model incorporates forward-looking behavior: the decisions of agents*  *depend on their expectations about the future behavior of the climate, which is itself influenced by those very decisions. Achieving consistency between agents' expectations and the climate trajectory thus requires solving for an interdependent equilibrium.*

*Successfully addressing these challenges lays the groundwork for using NorESM2–DIAM as a platform to explore the spatial and temporal dimensions of*

*climate–economy interactions, and to assess climate policy with a degree of geophysical and economic realism that is rare in existing IAMs. This platform contributes to a small but growing literature using dynamic, forward-looking, structural economic models to study the spatial effects of climate change (see, for example, Brock et al., 2014; Desmet and Rossi-Hansberg, 2015; Fried, 2022; Krusell and Smith, 2022; Rudik et al., 2021; Bilal and Rossi-Hansberg, 2023; Cruz and Rossi-Hansberg, 2024; Kubler, 2023; Kotlikoff et al., 2024).*

*Our approach to coupling an ESM and an IAM, embodied in NorESM2–DIAM, contrasts with the approach taken in iESM (Collins et al., 2015; Thornton et al., 2017; Calvin and Bond-Lamberty, 2018) and E3SM–GCAM (Di Vittorio et al., 2025), two other frameworks that couple an ESM and an IAM. The main difference is that both iESM and E3SM–GCAM couple an ESM with the Global Change Assessment Model (GCAM), a process-based rather than a cost-benefit IAM. Although both DIAM and GCAM are dynamic, recursive models, in DIAM agents make decisions taking into account the entire future time horizon, whereas GCAM solves for outcomes one step at a time, considering only the current state.*

*The two approaches also differ in spatial resolution and sectoral detail. GCAM represents multiple sectors—including energy, industry, transport, agriculture, and land use—but divides the world into only 14 (iESM) or 32 (E3SM-GCAM) socioeconomic regions. In contrast, NorESM2–DIAM contains only a single sector, focusing directly on gross domestic product (GDP), but at a very high degree of spatial resolution (1°×1° cells), enabling high-resolution analysis of the impacts of climate and weather on GDP and emissions.*

*Finally, the three models differ in how they represent climate–economy interactions. iESM and E3SM-GCAM exchange biogeochemical variables from the ESM to GCAM, whereas in our framework, temperature directly affects the economy through the productivity of labor. GCAM also explicitly represents agriculture and land use, allowing iESM and E3SM–GCAM to generate land-mediated feedbacks that are absent in NorESM–DIAM. Thus, although all three frameworks couple an IAM with an ESM, our approach employs a fundamentally different IAM, providing a complementary perspective to the two existing frameworks.*

2. Explanation of aggregate GDP deviations (Lines 600–602)

   The current explanation, that aggregate GDP deviations persist mainly due to spatially correlated temperatures and concentrated economic activity, is not entirely convincing to me. Spatial correlation in temperature does not necessarily translate to homogeneous economic responses. For example, the northern and southern United States likely respond differently to warming: although northern regions show greater temperature increases than southern

ones (Figure 11), they still experience gains in GDP per capita, whereas parts of the South show declines (Figure 12). This pattern may reflect differences in each region's position relative to the optimal temperature shown in Figure 3. I encourage the authors to revise this explanation.

We agree that the explanation needed to be fleshed out more. We now include a comparison with a simulation without spatial correlation (as simulated by the standalone model), which gives aggregated GDP deviations that are an order of magnitude smaller than in the coupled model with spatial correlation. Additionally, we have added a stylised example in the appendix to further demonstrate the role of spatial correlation in generating aggregate GDP fluctuations. (L623-633)

*Finally, as shown in Fig. 11 (b), global GDP itself experiences large fluctuations relative to its trend, about 1% in* magnitude. The patterns of spatial correlation in regional temperatures generated by NorESM2 play a key role in driving this variability in global GDP. To examine the role of these patterns, we simulated the behavior of the standalone model with regional temperature shocks drawn according to Eq. (2), with the parameters of the regional AR(1) processes calibrated to simulated data from NorESM2 as described in Sect. 4. In the standalone model, these shocks are assumed to be statistically independent across regions and therefore exhibit no spatial correlation by construction. In this case, fluctuations in global GDP are about 0.1%, an order of magnitude smaller than in the fully-coupled model. Failing to account for patterns of spatial correlation would therefore lead to a large understatement of volatility in global GDP.

*To gain further insight into the role of spatial correlation in generating aggregate fluctuations, Sect. A8 in Appendix A shows analytically, in a stylized model, that spatial correlation can amplify the size of these fluctuations under certain conditions that our calibrated model satisfies.*

Added in Appendix A1 (L708-709):

*Section A8 uses a stylized model to examine the role of spatial correlation in generating fluctuations in global aggregates.*

From Appendix A8 (L926-944)

**A8 Spatial correlation and aggregate fluctuations: a stylized model**

This section uses a stylized model that captures some of the key features of NorESM2–DIAM to show that spatial correlation in regional temperatures can amplify aggregate fluctuations under certain conditions that our calibrated model satisfies.

Consider a model in which regional temperatures are drawn from a jointly normal distribution at any point in time. Specifically, assume that

$$T = (T_1, \ldots, T_M)^\top \sim N(\overline{T}_i, \Sigma),$$

with $\Sigma_{ii} = \sigma^2$ and $\Sigma_{ij} = \rho\sigma^2$ for $i \neq j$, where $\rho \in [0,1]$. Define $\hat{T}_i \equiv T_i - T^*$ and $\mu_i \equiv \overline{T}_i - T^*$, where $T^*$ maximizes the damage function. Then $\hat{T}_i \sim N(\mu_i, \sigma^2)$ and $\text{corr}(T_i, T_j) = \rho$ for $i \neq j$, i.e., the deviation of regional temperature from the optimal temperature, $T^*$, is normally distributed with a region-specific mean, $\mu_i$, and a common variance, $\sigma^2$, across regions; and the correlation between temperature deviations in any pair of regions is equal to $\rho$.

Assume that the damage function $D(T_i) = \exp(-\kappa(T_i - T^*)^2)$, so that $D$ is symmetric around the optimal temperature (in NorESM2–DIAM, by contrast, the damage function is not quite symmetric).

Let each region be assigned a weight $w_i$, with

$$\sum_{i=1}^{M} w_i = 1.$$

and let $S$ be the weighted average of the logarithm of regional productivity:

$$S \equiv \sum_{i=1}^{M} w_i \log D(T_i) = -\lambda \sum_{i=1}^{M} w_i \hat{T}_i^2.$$

The variance of $S$ is then a measure of the volatility of aggregate productivity, one of the key drivers of volatility in global GDP. The question is how this variance varies as the correlation between regional temperatures increases. The variance is given by:

$$\text{Var}(S) = 2\sigma^2\lambda^2 \left[ \sigma^2 W_2 + 2M_2 + \sigma^2\rho^2(1 - W_2) + 2\rho(M_1^2 - M_2) \right],$$

where

$$M_1 \equiv \sum_{i=1}^{M} w_i\mu_i, \quad M_2 \equiv \sum_{i=1}^{M} w_i^2\mu_i^2, \quad W_2 \equiv \sum_{i=1}^{M} w_i^2.$$

This expression is an increasing function of $\rho$ over the entire range $[0,1]$ if and only if $M_1^2 - M_2 > 0$, or equivalently, if and only if $R \equiv M_1^2/M_2 > 1$.

This condition can be checked using different weighting schemes for the 16,826 distinct cells in NorESM2–DIAM, with the $\mu_i$s corresponding to the deviation of regional pre-industrial temperatures from the optimal temperature. Most relevant for global GDP is weights corresponding to regional GDP in 1990, in which case $R = 184.4$. Alternatively, for weights corresponding to regional population in 1990, $R = 722.5$. In both cases, therefore, the required condition is easily satisfied.

Although this simple model is quite stylized in that it does not correspond exactly to the behavior of NorESM2–DIAM, nonetheless these calculations do suggest that positive spatial correlation between regional temperatures amplifies aggregate fluctuations in NorESM2–DIAM. They are also consistent with the finding reported in Sect. 5.4 that the standalone model (in which spatial correlation is absent) produces much smaller aggregate fluctuations than the fully-coupled model.

**Minor Comments:**

1. Line 586: Please double-check "(see Fig. 4.5)". I cannot find Figure 4.5. Perhaps this refers to Section 4.5 instead?

Indeed, this should be "section" rather than "figure". This is now fixed. (L610)

*Second, average regional temperatures vary greatly across space, so that regions are located at very different points along the inverse U-shaped damage function (see* Sect.  *4.5) determining regional productivity.*

2. Figure 13(c): Some regional labels overlap and are difficult to read. I suggest adjusting the layout or font size to improve readability.

The labels are now adjusted so that they do not overlap. (Page 28)

[Figure]

3. Figure 5: There appears to be a sharp change between 2100 and 2120. Could the authors clarify the potential reason for this sharp change?

We have made the text clearer and added a clarification for this sharp change in the text (L451-465):

*… We assume that $\phi_t$ = 0 for $t \geq t_g$, after which point energy use is fully green.*

*…*

*This function is close to 1 when t=0 and declines slowly at first before accelerating, with H(10) = 0.99,  H(75) = 0.5, and H(140) = 0.01. …*

*…*

*To conserve on computation time, we run the fully-coupled model only until 2100, at which point we assume that energy use becomes fully green (i.e., we set $t_g$ = 111). We make this assumption so that, when computing decision rules using the standalone model, regional temperatures reach a steady state in 2100. The resulting small kink in the greening function has negligible effects on the quantitative results because annual emissions are already quite low by 2100 and, consequently, their impact on cumulative emissions is close to zero by that point.*